# Loss of Kat2a enhances transcriptional noise and depletes acute myeloid leukemia stem-like cells

Ana Filipa Domingues[1†‡§], Rashmi Kulkarni[1†#], George Giotopoulos[2,3], Shikha Gupta[1,4], Laura Vinnenberg[1], Liliana Arede[1,4], Elena Foerner[1], Mitra Khalili[1,5], Rita Romano Adao[1], Ayona Johns[6], Shengjiang Tan[2], Keti Zeka[1,4], Brian J Huntly[2,3], Sudhakaran Prabakaran[4,7], Cristina Pina[4,6]*

[1]Department of Haematology, University of Cambridge, NHS-BT Blood Donor Centre, Cambridge, United Kingdom; [2]Department of Haematology, University of Cambridge, Cambridge Institute for Medical Research, Cambridge, United Kingdom; [3]Wellcome Trust-Medical Research Council Cambridge Stem Cell Institute, Cambridge, United Kingdom; [4]Department of Genetics, University of Cambridge, Cambridge, United Kingdom; [5]Department of Medical Genetics and Molecular Medicine, School of Medicine, Zanjan University of Medical Sciences (ZUMS), Zanjan, Islamic Republic of Iran; [6]Division of Biosciences, College of Health and Life Sciences, Brunel University London, Uxbridge, United Kingdom; [7]Department of Biology, IISER, Pune, India

**\*For correspondence:**
cp533@cam.ac.uk;
cristina.pina@brunel.ac.uk

[†]These authors contributed equally to this work

**Present address:**
[‡]Haematological Cancer Genetics, Wellcome Trust Sanger Institute, Cambridge, United Kingdom; [§]Wellcome Trust-Medical Research Council Cambridge Stem Cell Institute, Cambridge, United Kingdom; [#]Department of Biochemistry, University of Cambridge, Cambridge, United Kingdom

**Competing interests:** The authors declare that no competing interests exist.

**Abstract** Acute Myeloid Leukemia (AML) is an aggressive hematological malignancy with abnormal progenitor self-renewal and defective white blood cell differentiation. Its pathogenesis comprises subversion of transcriptional regulation, through mutation and by hijacking normal chromatin regulation. Kat2a is a histone acetyltransferase central to promoter activity, that we recently associated with stability of pluripotency networks, and identified as a genetic vulnerability in AML. Through combined chromatin profiling and single-cell transcriptomics of a conditional knockout mouse, we demonstrate that Kat2a contributes to leukemia propagation through preservation of leukemia stem-like cells. Kat2a loss impacts transcription factor binding and reduces transcriptional burst frequency in a subset of gene promoters, generating enhanced variability of transcript levels. Destabilization of target programs shifts leukemia cell fate out of self-renewal into differentiation. We propose that control of transcriptional variability is central to leukemia stem-like cell propagation, and establish a paradigm exploitable in different tumors and distinct stages of cancer evolution.

## Introduction

Acute Myeloid Leukemia (AML) is the most prevalent leukemia in adults with a dismal prognosis of less than 30% 5 year survival (*Döhner et al., 2017*). It is a heterogeneous disease, clinically and pathologically, with common cellular themes of myeloid differentiation block, and recurrent molecular targeting of chromatin and transcriptional regulators. Effects on chromatin and transcription are reflected in the AML mutational spectrum (*Ley et al., 2013*), as well as through the implication of general chromatin regulators in AML pathogenesis, in the absence of specific mutation events (*Roe and Vakoc, 2013*). Specific examples of AML dependence on unmutated chromatin regulators include BRD4 (*Dawson et al., 2011*; *Zuber et al., 2011*), LSD1 (*Harris et al., 2012*) or DOT1L (*Bernt et al., 2011*; *Daigle et al., 2011*), their importance highlighted by the fact that chemical

**eLife digest** Less than 30% of patients with acute myeloid leukaemia – an aggressive cancer of the white blood cells – survive five years post-diagnosis. This disease disrupts the maturation of white blood cells, resulting in the accumulation of immature cells that multiply and survive but are incapable of completing their maturation process. Amongst these, a group of cancer cells known as leukemic stem cells is responsible for continually replenishing the leukaemia, thus perpetuating its growth.

Cancers develop when cells in the body acquire changes or mutations to their genetic makeup. The mutations that lead to acute myeloid leukaemia often affect the activity of genes known as epigenetic regulators. These genes regulate which proteins and other molecules cells make by controlling the way in which cells 'read' their genetic instructions.

The epigenetic regulator Kat2a is thought to 'tune' the frequency at which cells read their genetic instructions. This tuning mechanism decreases random fluctuations in the execution of the instructions cells receive to make proteins and other molecules. In turn, this helps to ensure that individual cells of the same type behave in a similar way, for example by keeping leukaemia cells in an immature state. Here, Domingues, Kulkarni et al. investigated whether interfering with Kat2a can make acute myeloid leukaemia less aggressive by allowing the immature white blood cells to mature.

Domingues, Kulkarni et al. genetically engineered mice to remove Kat2a from blood cells on demand and then inserted a mutation that causes acute myeloid leukaemia. The experiments showed that the loss of Kat2a delayed the development of leukaemia in the mice and progressively depleted leukaemia stem cells, causing the disease to become less aggressive. The results also showed that loss of Kat2a caused more fluctuations in how the white blood cells read their genetic code, which resulted in more variability in the molecules they produced and increased the tendency of the cells to mature.

These findings establish that loss of Kat2a causes leukaemia stem cells to mature and stop multiplying by untuning the frequency at which the cells read their genetic instructions. In the future, it may be possible to develop drugs that target human KAT2A to treat acute myeloid leukaemia.

inhibitors developed to target these regulators have progressed to clinical trials (*Gallipoli et al., 2015*). More recently, TAF12, a component of the basal transcription factor complex TFIID, was shown to be critical for AML cell maintenance through regulation of protein stability and activity of the transcription factor MYB (*Xu et al., 2018*).

In a recent CRISPR drop-out screen of genetic dependencies in AML, we have identified several members of the promoter-bound histone acetyl-transferase SAGA complex, including acetyl-transferase *KAT2A*, as being required for AML maintenance (*Tzelepis et al., 2016*). KAT2A was suggested to impact cell survival and differentiation status, but its precise molecular mechanisms of action remain to be elucidated, and it is unclear whether it is required in AML initiation, as well as maintenance. *Kat2a* is a mammalian orthologue of yeast histone acetyl-transferase *Gcn5*, and is required for H3K9 acetylation (H3K9ac) (*Jin et al., 2014*), a modification that fine-tunes, rather than initiates, locus-specific transcriptional activity. Kat2a is required for specification of mesodermal derivatives during early embryonic development (*Lin et al., 2007*; *Wang et al., 2018*), and for survival of neural stem and progenitor cells (*Martínez-Cerdeño et al., 2012*). Loss of *Kat2a* in the hematopoietic system from an early developmental stage did not grossly impact blood formation in vivo, but could promote terminal granulocyte differentiation in vitro, through relief of protein acetylation-dependent inactivation of transcription factor Cebpa (*Bararia et al., 2016*). Nevertheless, detailed testing of *Kat2a* contribution to hematopoietic stem and progenitor cell function is still lacking.

Yeast Gcn5 is a classical regulator of transcriptional noise (*Raser and O'Shea, 2004*), with deletion mutants enhancing cell-to-cell variability in gene expression measured across a range of locus fluorescence reporters (*Weinberger et al., 2012*). Transcriptional noise reflects the variability in the number of mRNA molecules produced from a given locus through time; snapshot studies of gene expression capture the same phenomenon as cell-to-cell transcriptional heterogeneity

(*Sanchez et al., 2013*). Transcriptional noise can result from the bursting nature of gene expression (*Chubb and Liverpool, 2010*). Most if not all loci, undergo bursts of transcriptional activity with characteristic frequency and size: burst frequency corresponds to the rate at which promoters become engaged in active transcription; burst size measures the number of mRNA molecules produced during each transcriptional burst (*Cai et al., 2006*). Both parameters contribute to mean gene expression, whereas transcriptional noise is more strictly dependent and shown to be anti-correlated with burst frequency (*Hornung et al., 2012*). In yeast, size and frequency of bursts are increased through histone acetylation of gene bodies and promoters, respectively (*Weinberger et al., 2012*).

In functional terms, transcriptional noise has been directly implicated as a mechanism of cell fate choice in yeast (*Blake et al., 2006*) and bacteria (*Süel et al., 2006*), and recurrently associated, albeit correlatively, with cell fate transitions in some mammalian systems (*Moris et al., 2016*). We had previously shown that normal transitions into hematopoietic lineage specification associate with cell-to-cell transcriptional heterogeneity (*Pina et al., 2012*; *Teles et al., 2013*). More recently, we have inhibited the activity of Kat2a in mouse embryonic stem cells, and observed an increase in transcriptional heterogeneity that impacted the stability of pluripotency with rewiring of correlation gene regulatory networks (GRNs) (*Moris et al., 2018*). Whilst we have not mechanistically linked enhanced heterogeneity with loss of pluripotency, we did observe propagation of variability of transcriptional levels through the GRNs, downstream of nodes with differential H3K9ac.

Cancer, and in particular leukemia, can be perceived as a pathogenic imbalance of cell fate choices, with maintained self-renewal and reduced exit into differentiation. We postulated that enhancing transcriptional variability in AML cells, as may be achieved through *Kat2a* depletion, would enhance the probability of differentiation cell fate transitions. By using a retroviral-delivered *MLL-AF9* model of AML in a conditional *Kat2a* knockout background, we show that loss of *Kat2a* depletes AML stem-like cells, imposing a mild delay to disease initiation and severely impairing AML propagation. At a molecular level, these changes are accompanied by specific loss of H3K9 acetylation at a subset of promoters, with reduced transcription factor binding and frequency of bursting, and associated variability in transcriptional levels. Affected loci encode for general mitochondria and nucleic acid metabolism, including translation, suggesting that they may contribute to cancer fate decisions more generally, rather than in a disease-specific manner.

## Results

### Inducible loss of Kat2a in adult mouse hematopoiesis

In order to investigate the impact of *Kat2a* loss on self-renewal vs. differentiation of AML stem-like cells (AML-SC), we generated an inducible conditional *Kat2a*$^{Fl/Fl}$ KO mouse model and transformed *Kat2a* excised (KO) and non-excised (WT) bone marrow (BM) cells through retroviral delivery of an *MLL-AF9* fusion transcript. This strategy allows for cellular and molecular investigation of *Kat2a* requirements during transformation, whilst minimizing putative confounding effects of acquired mutations on heterogeneity of transcription, such as might be observed in established human or mouse AML. The choice of a strong oncogenic event such as *MLL-AF9* (*Ley et al., 2013*; *Krivtsov et al., 2006*; *Somervaille and Cleary, 2006*) minimizes the need for cooperating genetic alterations. Similarly, the option for an inducible conditional knockout may reduce compensatory effects against *Kat2a* loss. Specifically, we crossed *Kat2a*$^{Fl/Fl}$ C57Bl/6 mice (*Lin et al., 2008*) on the interferon response-inducible *Mx1-Cre*$^{+/-}$ background, and generated a stable mouse line homozygous for the *Flox* allele. *Mx1-Cre*-positive (KO) and *Mx1-Cre*-negative (WT) mice were compared across all experiments (*Figure 1—figure supplement 1A*), with locus excision obtained through treatment of experimental and control mice with intra-peritoneal polyinosylic-polycytidylic (pIpC) acid (*Chan et al., 2011*). Excision was tested 4–6 weeks after injection and consistently achieved values greater than 80% in stem and progenitor cell compartments (*Figure 1—figure supplement 1B*), reflected in a profound loss of gene expression, including amongst myeloid-biased (LMPP) and committed (GMP) progenitors critical for AML initiation (*Goardon et al., 2011*; *Figure 1—figure supplement 1C*). Locus excision generates an in-frame product joining the first two and the last exons and is transcribed (*Figure 1—figure supplement 1D*) but does not code for acetyl-transferase activity or other functional domains (*Figure 1—figure supplement 1A*). *Kat2a* was dispensable for normal mouse hematopoiesis, with minimal transient effects on the number (*Figure 1—figure supplement*

1E) and in vitro activity (*Figure 1—figure supplement 1F*) of KO hematopoietic stem cells (HSC). These effects were not sustained through aging (*Figure 1—figure supplement 1G*), and, crucially, did not affect HSC function in vivo, measured by long-term reconstitution of irradiated recipients through transplantation (*Figure 1—figure supplement 1H*).

## Kat2a loss impairs establishment of MLL-AF9 leukemia

Having verified that *Kat2a* deletion does not perturb normal hematopoiesis and thus preserves candidate progenitor cells-of-origin for leukemia transformation, we used a retroviral delivery system to express the *MLL-AF9* leukemia fusion in progenitor-enriched, lineage-depleted (Lin-) WT and KO BM cells. Cells were transformed in vitro through serial re-plating in semi-solid medium-based colony-forming assays, with similar efficiency for both genotypes (*Figure 1A*). Importantly, the level of locus excision was mildly increased during transformation (*Figure 1B*), indicating that loss of *Kat2a* does not impede the initial selection of leukemia-transformed clones. Continued re-plating revealed that *Kat2a* KO affected the type of colonies produced, with a shift from compact or type I (*Johnson et al., 2003*) colonies (*Figure 1C–D*) with Kit$^+$ progenitor-like cells (*Figure 1E* and *Figure 1—figure supplement 2A–B*), to mixed, or type II (*Johnson et al., 2003*) colonies (*Figure 1F*), with a characteristic halo (*Figure 1G* and *Figure 1—figure supplement 2C*) of more differentiated cells and a corresponding reduction in Kit-positive cells, which have higher levels of the differentiation marker Mac1/CD11b (*Figure 1H* and *Figure 1—figure supplement 2D*). Compatible with the serial re-plating experiments, cell lines established from *MLL-AF9*-transformed cell lines of both *Kat2a* WT and KO genotypes exhibited a relative gain in mixed colonies (*Figure 1—figure supplement 2E*), with higher levels of Mac1 (*Figure 1—figure supplement 2F*). Taken together, the data suggest that *Kat2a* loss is permissive to *MLL-AF9*-driven transformation, but alters the balance between in vitro self-renewal and differentiation, favoring the latter. We tested these observations in vivo by monitoring leukemia development in mice that received WT and KO Lin- BM cells transduced for 2 days with retrovirus encoding the *MLL-AF9* oncogenic fusion. *Kat2a* KO recipient animals had a moderate advantage in survival (*Figure 2A*), suggesting a protracted development of leukemia. Leukemias were nevertheless depleted of *Kat2a* expression (*Figure 2B*), thus excluding selection of escapee cells. WT and KO leukemic animals had similar disease burdens at the point of culling, as measured by organ infiltration (*Figure 2—figure supplement 1A*) and peripheral blood counts (*Figure 2—figure supplement 1B*). In contrast with in vitro transformation, we did not observe gross changes to the balance between progenitor and differentiated cells in the resulting WT vs KO leukemias, which had preservation of phenotypic leukemia stem-like cells (LSC), defined by the Lin-Kit+Sca1-CD34+CD16/32+ GMP surface phenotype (L-GMP) (*Figure 2C*). We reckoned that loss of *Kat2a* affected the probability of leukemia development through dysregulation of transformed cells, and sought to probe this hypothesis by investigating the transcriptional programs of WT and KO leukemias at the single-cell level.

## Kat2a KO leukemias have increased transcriptional variability

To this end, we pooled samples from 4 to 5 primary leukemias of each genotype, sorted live GFP+ cells reporting the presence of the *MLL-AF9* fusion, and successfully sequenced over 4000 cells *Kat2a* WT and KO cells each, for a total of 13166 transcripts, using a high-throughput 10X Genomics platform (*Zheng et al., 2017*). Basic measures of gene alignment and quality control are summarized in *Supplementary file 1*. In order to minimize the confounding effect of rarely or very low expression genes, we filtered out transcripts detected in less than 10% of all cells, and only included cells where a minimum of 500 different transcripts were detected. These filtering criteria match those recently used in analysis of heterogeneity of hematological malignancies using a similar 10X Genomics platform (*Pastore et al., 2019*) and were deliberately chosen for reproducibility. We thus focused the analysis on what we designated Robust gene set. These were a total of 2588 genes (*Supplementary file 2*), sampled from 7360 cells (3835 WT, 3525 KO), which captured general metabolic, biosynthetic and regulatory ontologies, as well as hematopoietic-specific categories, for example 'myeloid cell differentiation', indicating broad representation of transcriptional programs within the dataset (*Supplementary file 3*). We assessed gene expression using the D3E algorithm (*Delmans and Hemberg, 2016*), which makes implicit use of the single-cell nature of the data to extract information about differential dynamics of transcription, in addition to determining

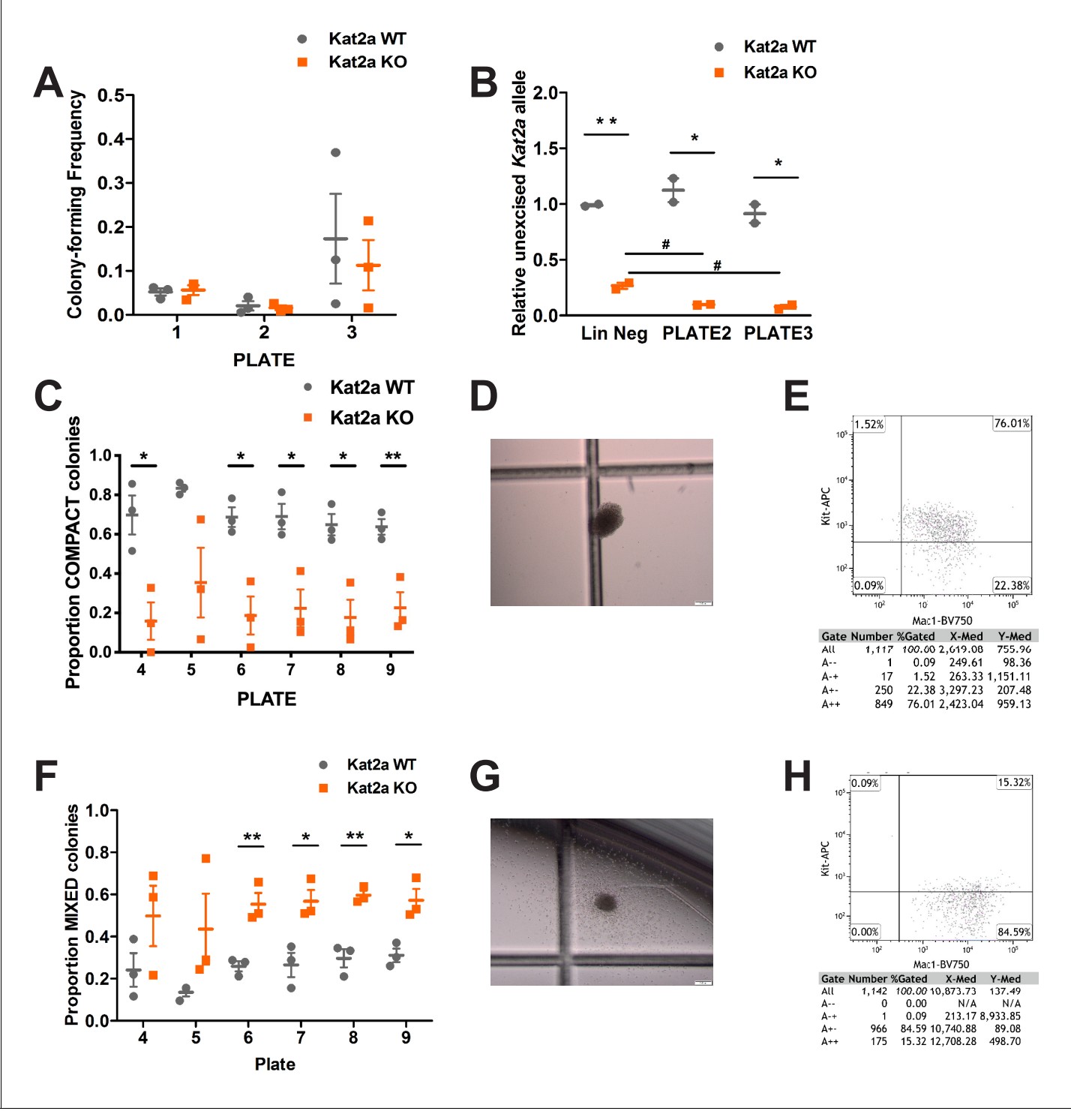

**Figure 1.** Conditional knockout of *Kat2a* promotes differentiation of *MLL-AF9*-transformed cells in vitro. (**A**) Serial re-plating of colony-forming cell (CFC) assays of MLL-AF9 transformed cells, mean ± SEM, n = 3. (**B**) Excision efficiency was evaluated by qPCR during re-plating of *MLL-AF9* transformed cells, mean ± SEM, n = 2–3, *p<0.01 and **p<0.001. (**C**) Proportion of Compact-type colonies in *MLL-AF9* transformed cells on *Kat2a* WT or KO background, mean ± SEM, n = 3, *p<0.01 and **p<0.001. (**D**) Representative photograph of a Compact-type colony. (**E**) Flow cytometry analysis of the colony in (**D**). (**F**) Proportion of Mixed-type colonies in *MLL-AF9* transformed cells on *Kat2a* WT or KO background, mean ± SEM, n = 3, *p<0.01 and **p<0.001. (**G**) Representative photograph of a Mixed-type colony. (**H**) Flow cytometry analysis of the colony in (**G**). Two-tailed t-test was performed in (**A**), (**B**), (**C**) and (**F**).

The online version of this article includes the following source data and figure supplement(s) for figure 1:

*Figure 1 continued on next page*

*Figure 1 continued*

**Figure supplement 1.** Loss of *Kat2a* does not affect normal hematopoiesis.
**Figure supplement 1—source data 1.** Colony-forming assays of *Kat2a* WT and KO stem and progenitor cells.
**Figure supplement 2.** Loss of *Kat2a* promotes differentiation of MLL-AF9 leukemia cells in vitro.
**Figure supplement 2—source data 1.** Differential colony counts from liquid cultures of *MLL-AF9* transformed *Kat2a* WT and KO cells.

changes in average gene expression. The algorithm views single-cell measurements as multiple observations of the same or closely-related cells and fits a two-state promoter model that interprets gene expression measurements in terms of frequency and size of bursts of promoter activity (*Figure 3—figure supplement 1A*). We used a multiple linear regression model to verify the contribution of burst size and frequency to average gene expression and gene expression CV in our single-cell RNA-seq data (*Figure 3—figure supplement 1B*). In line with other studies (*Hornung et al., 2012*), we observed that both bursting parameters contributed to mean expression to similar extents, but there was a greater contribution of burst frequency to CV. Differential gene expression analysis between *Kat2a* WT and KO primary leukemic cells revealed mild albeit significant changes in average gene expression (*Figure 3A*; median difference −0.03), which were of down-regulation upon *Kat2a* excision, as might be anticipated from loss of a histone acetyl-transferase. Compatible with the proposed role of Kat2a in regulating transcriptional noise, we observed a significant increase in gene expression variability as measured by coefficient of variation (CV = standard deviation/mean) (*Figure 3B*; median difference 0.24) in *Kat2a* KO cells which was apparent at all levels of gene expression (*Figure 3C*). The increase in gene expression CV associates with greater cell-to-cell dispersion (*Mohammed et al., 2017*), in other words reduced cell-to-cell correlation in transcript levels, amongst *Kat2a* KO leukemia cells (*Figure 3D*). The increased dispersion of transcriptional variability can be attributable to a change in burst frequency (*Figure 3E*; median difference −1.15), but not burst size (*Figure 3F*). This has also been observed for the *Kat2a* yeast orthologue *Gcn5* in modulating transcriptional noise (*Weinberger et al., 2012*).

## Increased transcriptional variability associates with loss of Kat2a KO leukemia stem-like cells

There have been several reports (*Mohammed et al., 2017*; *Antolović et al., 2017*; *Blake et al., 2006*; *Chambers et al., 2007*; *Chang et al., 2008*; *Reynolds et al., 2012*), including our own (*Pina et al., 2012*), that suggest an association between variability in gene expression and probability of cell fate change, although most of the data remains correlative, at least in mammalian systems. Our recent analysis of Kat2a inhibition in mouse ES cells (*Moris et al., 2018*) is compatible with this view, suggesting that the observed enhanced transcriptional variability may promote exit from pluripotency through disruption of gene regulatory networks. In this light, we asked if its enhancement in *Kat2a* KO leukemias resulted in an imbalance of self-renewal vs differentiation states that could lead to the observed delay in leukemia progression (*Figure 2A*). We made use of the RACE-ID algorithm (*Grün et al., 2016*) to cluster the combined 7360 WT and KO cells filtered as displaying Robust gene expression, and optimally separated them into 12 clusters on the basis of the 500 most highly variable genes in each genotype (*Figure 4A*). Despite the occupancy of a broadly similar transcriptional space, the genotypes had differential cluster-association patterns, with some clusters, namely 7, 11 and 12, which together comprise 22.9% of WT cells, being relatively underrepresented amongst *Kat2a* KO cells (respectively 0.5, 0.2 and 0.5 of *Kat2a* WT) (*Figure 4—figure supplement 1A*). Other clusters were over-represented amongst *Kat2a* KO cells but to a lesser extent, with only 2 over 1.5-fold (cluster 4–1.7-fold, 3.3% of WT cells; cluster 6–1.5-fold, 8.5% of WT cells), causing us to focus on the *Kat2a* KO-depleted clusters. Unsupervised alignment of the 12 clusters along a putative differentiation trajectory, shows that the underrepresented clusters lie at its undifferentiated end (*Figure 4B*). This is enriched for gene expression signatures associated with MLL self-renewal (*Figure 4—figure supplement 1B*), suggesting a depletion of cells with stem-like characteristics. Overall, gene expression programs at the start of the trajectory were similar between clusters 2, 4 and 7, with few genes unique to individual clusters thus preventing exploration of gene ontologies. There were nevertheless subtle differences in the functional categories enriched amongst differentially expressed genes between genotypes in each of the clusters, namely an association of

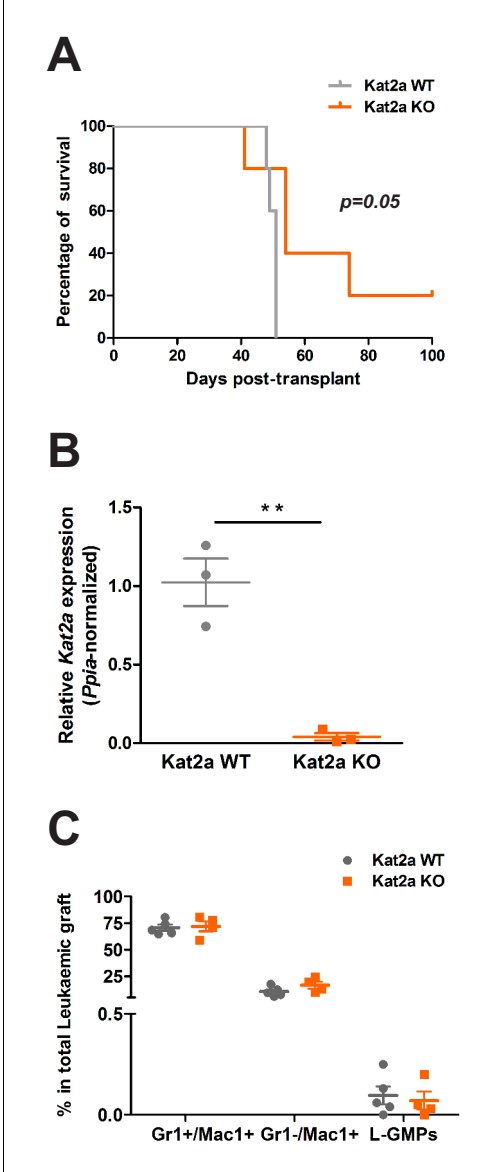

**Figure 2.** *Kat2a* loss impairs establishment of *MLL-AF9* leukemia in vivo. (A) Survival curve of animals transplanted with *MLL-AF9* transformed *Kat2a* WT or KO cells. N = 5 animals/genotype; log rank test, p=0.05. (B) Relative expression (quantitative RT-PCR) of *Kat2a* in *MLL-AF9* primary leukemia BM cells from *Kat2a* WT and KO backgrounds, mean ± SEM, n = 3, **p<0.001, 2-tailed t-test. (C) Flow cytometry analysis of BM cellularity of primary *Kat2a* WT or KO leukemias: shown are late (Mac1⁺Gr1⁺) and early (Mac1⁺Gr1⁻) differentiated populations and Gr1⁻Mac1⁻cKit⁺Sca1⁻CD34⁺FcgR⁺ candidate stem-like L-GMP cells, mean ± SEM, n = 4–5; 2-tailed t-test performed.

The online version of this article includes the following figure supplement(s) for figure 2:

**Figure supplement 1.** Primary *Kat2a* WT and KO *MLL-AF9* leukemias have similar disease burden.

categories associated with apoptosis in clusters 2 and 4, which were absent from cluster 7 (*Supplementary file 4*). The STEM-ID algorithm builds on RACE-ID to define a 'stemness' score that has successfully identified previously elusive stem cell populations in mouse pancreas (*Grün et al., 2016*) and cellular hierarchies in human liver (*Aizarani et al., 2019*). It postulates that stem cells exhibit a multitude of incipient lineage-affiliated programs (high information entropy), which are shared (high connectivity) with more differentiated cells, and attributes a 'stemness' score to each cluster as the product of entropy and number of links (*Figure 4—figure supplement 1C*). Indeed, cluster seven scores as the most stem-like transcriptional state, suggesting that *Kat2a* KO leukemias may be depleted of functional, if not surface phenotype (*Figure 2C*), LSC. We tested this by pooling equal numbers of cells from all primary WT or all primary KO leukemias in *Figure 2A*, and transplanting them into secondary recipients at limiting cell doses. Despite the equivalent number of phenotypic LSC in the primary leukemias of both genotypes (*Figure 2C*), recipients of *Kat2a* KO leukemia cells failed to develop leukemia at the lowest cell doses and had a dramatically reduced frequency of functional LSC (*Figure 4C–D*), indicating that a requirement for Kat2a in leukemia self-renewal and/or propagation. Significantly, the level of Kat2a gene expression knockout was profound and retained in secondary leukemias, similar to the primary leukemias they originated from (*Figure 4—figure supplement 1D*). In agreement, the frequency of *Kat2a* KO cells escaping the excision was negligible amongst phenotypically undifferentiated Lin-Kit+Sca1-CD16/32+ cells (*Figure 4—figure supplement 1E*). This confirmation of a maintained and profound gene expression knockout with minimum contribution from undeleted cells is particularly important, as the persistence of a rearranged transcript fusing exons 1, 2 and 18 (*Figure 1—figure supplement 1D*) prevents knockout quantification in the 3'-biased 10X Genomics platform. In contrast, the Taqman assay and the nested primers in the PCR analyses (*Figure 4—figure supplement 1D–E*) are directed at exons 6–7 and 11–13, respectively, which capture the excised region. Thus, cellular and molecular analyses, including inspection of transcriptional variability, are likely minimally affected by the presence of cells escaping *Kat2a* gene deletion, and should truly reflect the effects of *Kat2a* loss. Secondary leukemias, like primary leukemias, did not have a reduction in surface phenotypic LSC (L-GMP, *Figure 4E*),

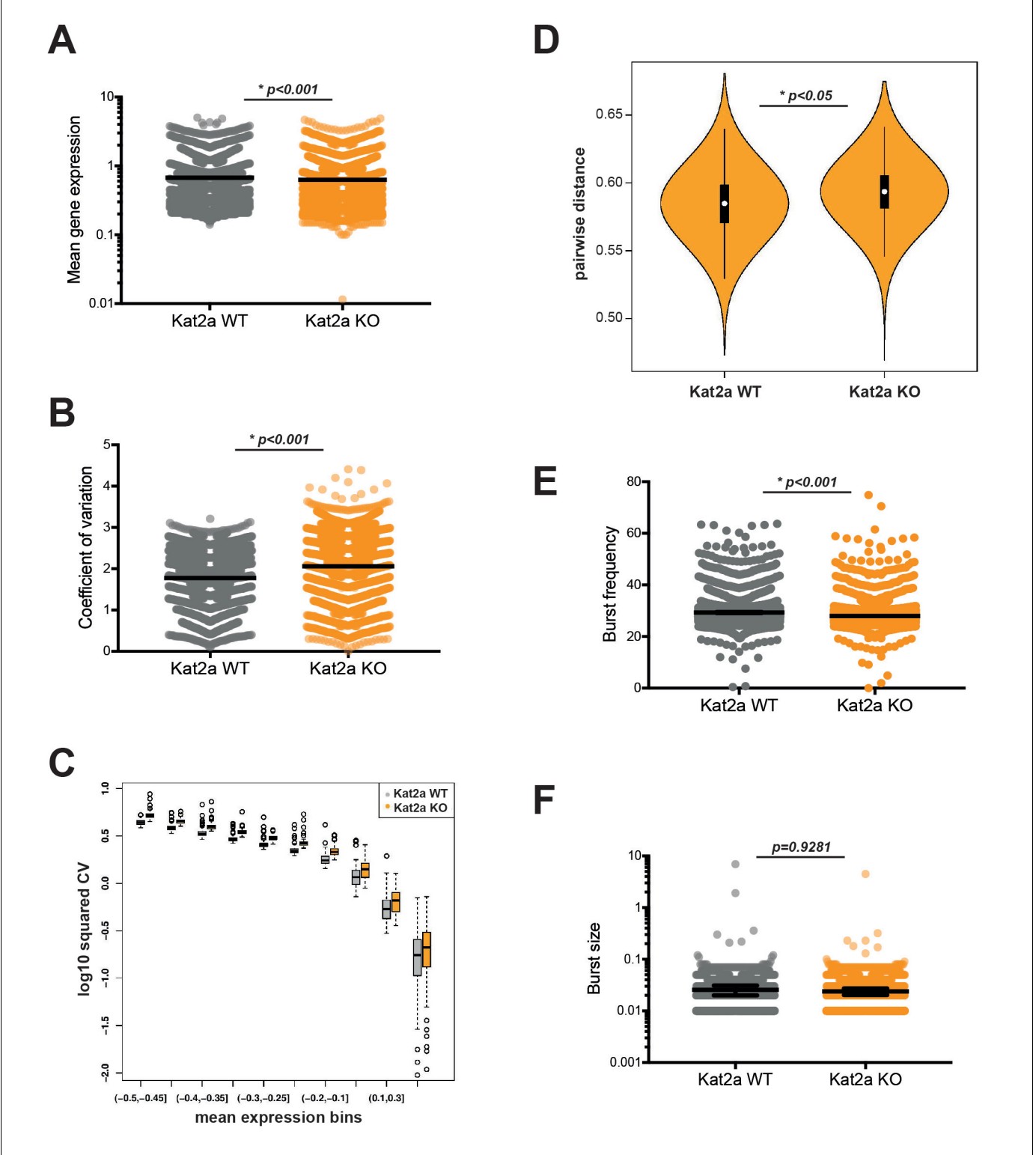

**Figure 3.** Loss of *Kat2a* increases transcriptional heterogeneity of primary *MLL-AF9* leukemias. (**A**) Mean gene expression levels in *Kat2a* WT and KO primary leukemia cells. Median and 95% CI of mean gene expression levels for the 2588 genes in the Robust gene set, across 7360 cells Kolmogorov-Smirnov (KS) non-parametric test, p-value<0.01. (**B**) Gene expression CV in *Kat2a* WT and KO primary leukemia cells. Data as in (**A**) : KS non-parametric test, p-value<0.01. (**C**) Binned gene expression CV across the distribution of gene expression averages for *Kat2a* WT and KO primary leukemia cells, KS non-parametric test, p-value<0.05 for all bins. (**D**) Pair-wise distance measure between any two genes across *Kat2a*-WT and KO primary leukemia cells. The top 500 most variable genes in the Robust gene set for each genotype, as determined by distance to the mean CV, were used, as previously

*Figure 3 continued on next page*

*Figure 3 continued*

described (*Mohammed et al., 2017*). Welch t-test for comparison of means *p-value<0.01. (E) Distribution of burst frequencies for the Robust gene set in *Kat2a* WT and KO primary leukemias, as calculated by the D3E algorithm. KS non-parametric test, * p-value<0.0001. (F) Distribution of burst sizes for the Robust gene set in *Kat2a* WT and KO primary leukemias, as calculated by the D3E algorithm. KS non-parametric test, non-significant.

The online version of this article includes the following source data, source code and figure supplement(s) for figure 3:

**Source code 1.** Binned CV analysis - R-language code and input matrix, source code for *Figure 3C*.
**Source code 2.** Pairwise distance measure - R-language code, source code for *Figure 3D*.
**Source data 1.** D3E output analysis of the Robust gene set with annotation of Kat2a acetylation targets.
**Figure supplement 1.** Differential transcriptional heterogeneity in *Kat2a* WT and KO *MLL-AF9* primary leukemias.
**Figure supplement 1—source code 1.** Multiple linear regression analysis - R-language code and input data, source code for *Figure 3—figure supplement 1*.

putatively highlighting a dissociation between primitive surface phenotype and function, or highlighting the existence of multiple stem-like states, as suggested by cluster-specific depletion at the undifferentiated end of the *Kat2a* KO leukemia trajectory. Nevertheless, secondary leukemias displayed an increased proportion of early (Mac1+Gr1-), but not late (Mac1+Gr1+) differentiated myelo-monocytic cells (*Figure 4E*), suggesting that loss of *Kat2a* may perturb self-renewal and differentiation progression at multiple levels. Accordingly, we observed reduction of transcriptional burst frequency in KO cells to be prevalent across clusters, albeit slightly more marked at the undifferentiated end (*Figure 4—figure supplement 1F*). The exception are the central clusters 11 and 12, which have minimal representation in *Kat2a* KO leukemia (*Figure 4—figure supplement 1A*). Also, separate inspection of the differentiation trajectories for *Kat2a* WT (*Figure 4—figure supplement 2A–B*) and *Kat2a* KO cells (*Figure 4—figure supplement 2C–D*) suggests that the almost linear developmental relationship between WT leukemia cells is replaced by multiple branching decisions upon *Kat2a* loss, an observation also captured by a second pseudo-time trajectory algorithm (*Figure 4—figure supplement 2E–F*). In summary, loss of *Kat2a* depletes functional LSC and alters cellular hierarchies within *MLL-AF9* leukemia. This is likely achieved through perturbation of promoter activity and we sought to define the genes and pathways directly regulated by Kat2a.

## Kat2a regulates transcription factor binding and bursting activity of promoters

We defined Kat2a regulatory targets by chromatin immunoprecipitation followed by next generation-sequencing (ChIP-seq). Kat2a is a histone acetyl-transferase required for deposition of the activating H3K9 acetylation (ac) mark (*Jin et al., 2014*) and capable of catalyzing multiple acetyl-modifications (*Kuo and Andrews, 2013*) at promoters and at enhancers (*Krebs et al., 2011*). We identified promoters as H3K4 tri-methyl (me3) peaks (*Figure 5—figure supplement 1A*) and enhancers with H3K4 mono-methyl (me1)- enriched regions (*Figure 5—figure supplement 1B*) and inspected the respective pattern of distribution of H3K9ac in pooled MLL-AF9 primary leukemias initiated by *Kat2a* KO or WT cells. Experiments were performed in duplicate, with good overlap between replicates (*Figure 5—figure supplement 1C–D*). We included analysis of another acetylation activation mark, H3K27ac, which associates with active enhancers (*Creyghton et al., 2010*), but has not been specifically linked to Kat2a activity. Inspection of peak overlaps between methylation and acetylation marks indicated a specific depletion of H3K9ac binding at promoters in regions devoid of concomitant H3K27ac activation mark (*Figure 5A*). Conversely, H3K9ac was mildly increased at candidate active enhancer regions marked by the presence of H3K27ac (*Figure 5—figure supplement 1E*) suggesting a possible pattern of imbalance of H3K9ac regulation between promoters and enhancers. Guided by these preliminary observations, we focused on those promoter peaks with unique loss of H3K9ac upon *Kat2a* depletion, and used the ENCODE database (*Auerbach et al., 2013*) to confirm enriched experimental binding of Kat2a (aka GCN5) in other model systems (*Figure 5B* and *Supplementary file 5*). Genes associated with differentially-acetylated promoter peaks represented general nucleic acid and mitochondrial metabolic categories (*Supplementary file 6*). Somewhat surprisingly, the list of target promoters failed to include most MLL-AF9 ChIP targets (*Bernt et al., 2011*) with the leukemia self-renewal associated *Hox* gene signature, namely *Hoxa9*, *Hoxa10* and *Meis1*, escaping Kat2a-dependent promoter acetylation control. This suggests that Kat2a-mediated maintenance of LSC may be achieved through general, rather

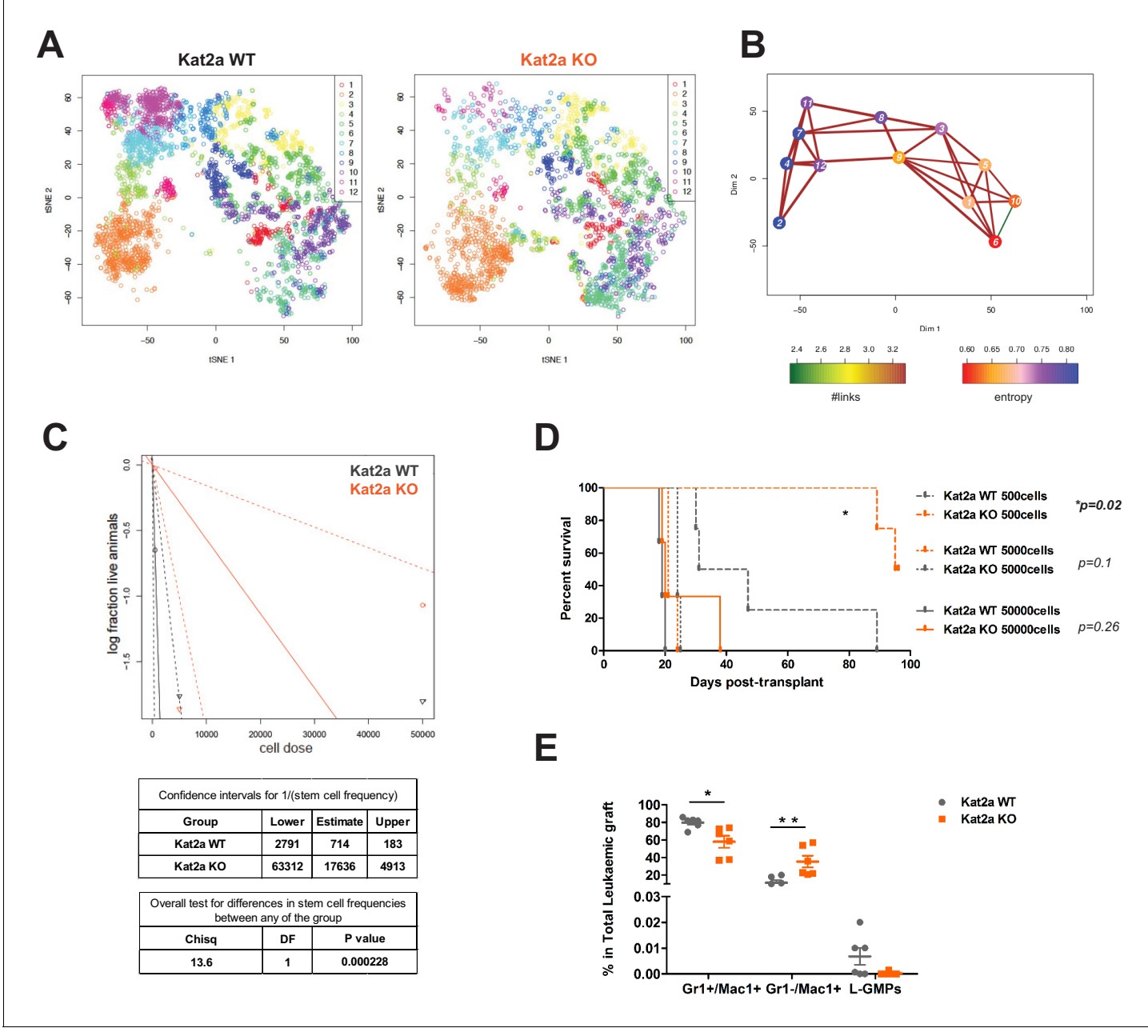

**Figure 4.** *Kat2a* loss depletes functional *MLL-AF9* leukemia stem-like cells. (A) t-SNE plot of single-cell RNA-seq data for *Kat2a* WT (left) and KO (right) primary leukemic cells. RACE-ID K-means clustering was used to classify cells from *Kat2a* WT and KO primary leukemias in combination, on the basis of the expression of the most highly variable genes from each genotype as defined in *Figure 2D*. Clusters are color-coded and cells of each genotype were displayed separately for easier appreciation of their non-overlapping transcriptional spaces. (B) STEM-ID trajectory plot of analysis in (A) representing combined measures of information entropy and cluster connectivity strength; clusters as in (A). (C) Extreme Limiting Dilution Analysis (ELDA *Hu and Smyth, 2009*) of leukemia-initiating cell frequency in *Kat2a* WT and KO primary leukemias. Primary leukemias of each genotype were pooled (WT-5; KO-4) and transplanted as 50K, 5K and 500 cell doses into 3–4 animals/dose group. (D) Survival curve of secondary recipients of *MLL-AF9* leukemic cells from *Kat2a* WT and KO backgrounds; data as in (C). Log rank test for difference in survival, n = 3–4/per dose group. 50 K cells p=0.26, 5 K cells p=0.1, 500 cells p=0.02. (E) Flow cytometry analysis of BM cells from secondary *Kat2a* WT and KO leukemia transplant recipients (50K and 5 K cells). Cell compartments as in *Figure 2C*; n = 6; mean ± SEM, 2-tailed t-test, **p<0.001, *p<0.01, and L-GMPs p=0.07.

The online version of this article includes the following source data and figure supplement(s) for figure 4:

**Source code 1.** tSNE plot of single-cell RNA-seq data - R-language code and individual cell coordinates with respective cluster ID, source code for *Figure 4A*.

**Figure supplement 1.** *Kat2a* WT and KO *MLL-AF9* primary leukemias have distinct cluster composition and organization.

**Figure supplement 2.** *Kat2a* WT and KO *MLL-AF9* primary leukemias have unique differentiation trajectories.

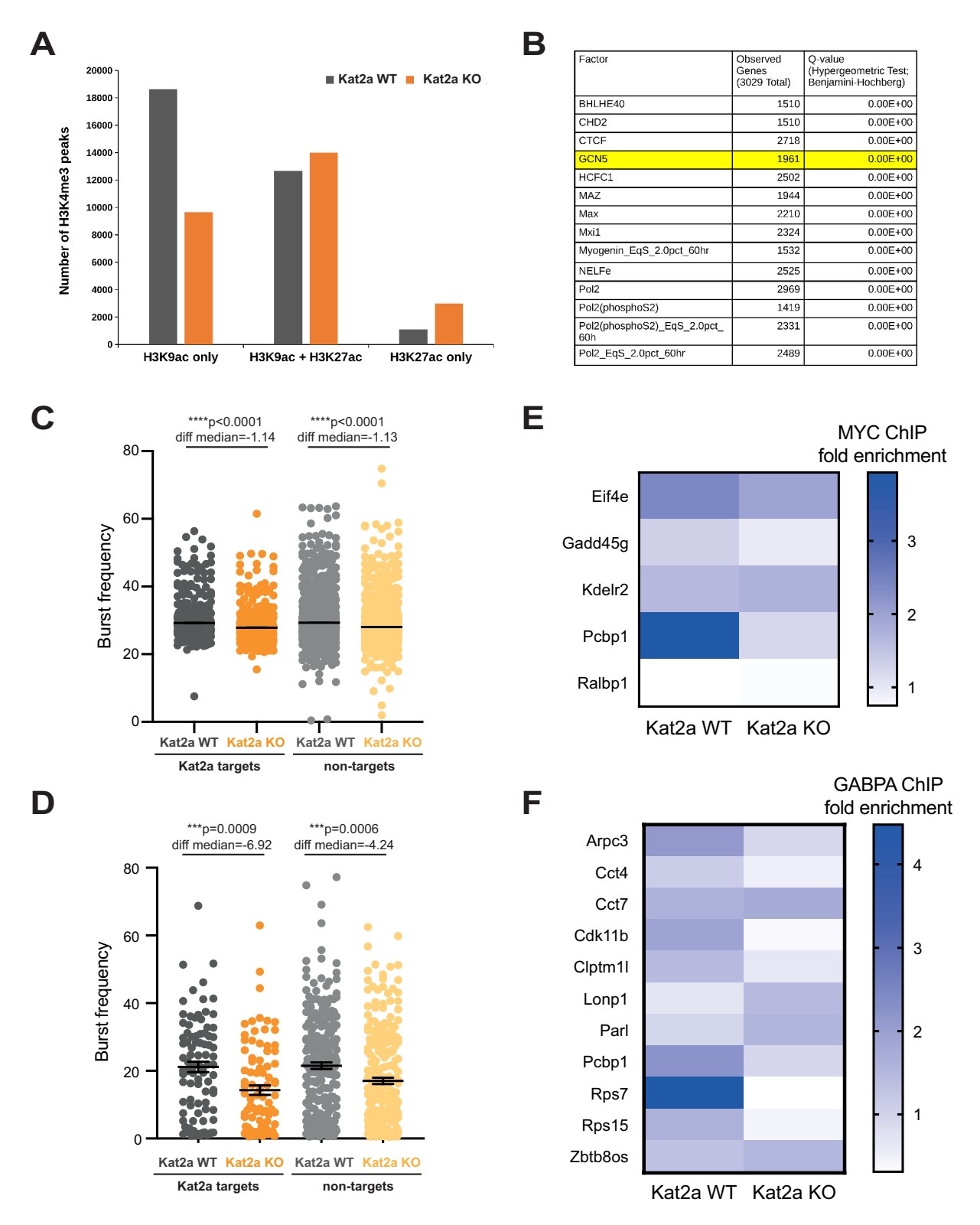

**Figure 5.** Loss of *Kat2a* depletes H3K9ac and perturbs transcription factor-binding in a subset of regulatory gene promoters. (**A**) Quantification of H3K9ac and H3K27ac ChIPseq peaks at H3K4me3 gene promoters in *Kat2a* WT and KO primary *MLL-AF9* leukemias. (**B**) Top ENCODE ChIP-seq Significance Tool enrichments for H3K9ac-positive promoters exclusive to *Kat2a* WT primary *MLL-AF9* leukemia cells. These promoters constitute Kat2a acetylation targets. (**C**) Distribution of burst frequencies for Kat2a acetylation targets vs. non-targets within the 2588-Robust gene set. Values as

*Figure 5 continued on next page*

Figure 5 continued

calculated by D3E with 2585 genes called differential. Mann-Whitney non-parametric test for comparison of rank medians; p<0.0001 for *Kat2a* WT vs. KO comparisons. WT vs KO median rank differential non-significant. (D) Distribution of burst frequencies for Kat2a acetylation targets vs. non-targets amongst cells in cluster 7; 857 genes considered as used in RACE-ID. Burst frequencies were calculated by D3E, with 332 genes called differential. Mann-Whitney non-parametric test for comparison of rank medians; p<0.001 for *Kat2a* WT vs. KO comparisons. (E) ChIP-quantitative PCR analysis of Myc binding in selected Kat2a acetylation target promoter peaks; mean values for 2–4 independent experiments using pooled BM or Spleen of *Kat2a* WT vs KO *MLL-AF9* secondary leukemias. Mean enrichments: WT = 2.158, KO = 1.357, 95% CI of difference 0.01273 to 1.589; 2-way ANOVA p<0.05 for genotype contribution. (F) ChIP-quantitative PCR analysis of Gabpa binding in selected Kat2a acetylation target promoter peaks; mean values for two independent experiments using pooled BM of *Kat2a* WT vs KO *MLL-AF9* secondary leukemias. Mean enrichments: WT = 1.775, KO = 0.9609, 95% CI of difference 0.2640 to 1.364; 2-way ANOVA p<0.01 for genotype contribution.

The online version of this article includes the following source data and figure supplement(s) for figure 5:

**Source data 1.** D3E output analysis of cluster seven with annotation of Kat2a acetylation targets.
**Figure supplement 1.** ChIP identification of regulatory regions in primary *MLL-AF9* leukemia.
**Figure supplement 2.** Loss of Kat2a affects transcription factor binding in a subset of loci.

than leukemia-specific, regulatory programs, with putative implications for other leukemic and non-leukemic malignancies. Having previously observed that loss of *Kat2a* associates with a decrease in frequency of transcriptional bursting (*Figure 3E*), we asked if this was more evident for genes dependent on Kat2a for promoter H3K9ac. Global inspection of the Robust gene set across all cells showed a similar reduction in burst frequency for Kat2a acetylation targets and non-targets (*Figure 5C*). However, H3K9ac target genes had uniquely preserved mean gene expression levels (95% CI KO-WT means = −0.1118 to 0.01303, Welch t-test p=0.12 for targets vs. −0.0894 to −0.0016, p=0.04 for non-targets), with increased CV upon Kat2a KO (both groups p<0.0001), suggesting a specific impact of Kat2a on variability of transcription. Nevertheless, global analysis of transcriptional burst frequency failed to produce a specific association with Kat2a acetylation targets, which could be due to a confounding effect of mixing cells at different points in the differentiation trajectory, for which locus-specific regulation might differ, or indeed a cascading of transcriptional consequences along gene regulatory networks, as suggested by our analysis of mouse ES cells where Kat2a activity was inhibited (*Moris et al., 2018*). Alternatively, the lack of association could reflect the fact that our subset of acetylation targets captures absolute, but not relative losses of promoter H3K9ac upon *Kat2a* KO, as a result of the necessarily limited replication of ChIP analysis of primary leukemia cells. To avoid potential confounding effects of cellular heterogeneity on determination of transcriptional parameters, we repeated the analysis exclusively on stem-like cluster 7 cells. Indeed, by using a more homogeneous group of cells, we could observe that the reduction in frequency of bursting in *Kat2a* KO cells was significantly more marked for H3K9ac target genes (*Figure 5D*; Mann-Whitney test for difference in burst frequency, Fburst, defined as (FburstKO-FBurstWT)/FburstWT p=0.0155), indeed suggesting that Kat2a acts by regulating bursting activity at target promoters. Transcriptional variability has been associated with density of transcription factor binding. As Kat2a is known to interact with transcription factors, including the oncogene Myc (*Wang et al., 2018*; *Hirsch et al., 2015*), to regulate common downstream gene expression programs, we asked if loss of Kat2a reduced transcription factor binding at target H3K9ac peaks. Analysis of H3K9ac targets using the ENCODE ChIP-Seq significance tool suggested binding by Myc in other experimental systems (*Supplementary file 3*), in addition to general transcription and pause-release factors. It also showed increased binding by the respiratory chain regulator Gabpa (*Ongwijitwat and Wong-Riley, 2005*) (Nrf2 in the ENCODE database), which has been described as a regulator of normal hematopoietic and Chronic Myeloid Leukemia (CML) stem cells (*Manukjan et al., 2016*; *Yang et al., 2013*). We performed ChIP-qPCR analysis of Myc and Gabpa binding at promoter peaks depleted of H3K9ac binding in *Kat2a* KO leukemias. We selected candidate Myc and Gabpa target peaks based on experimental DNA occupancy by the transcription factors (TF) across different mouse cell types in the ENCODE database. Q-PCR primers were designed under the respective H3K9ac peak, for analysis of pooled mouse *MLL-AF9* secondary leukemia samples of each genotype. We confirmed that the transcription factors (TF) Myc and Gabpa did indeed bind at most of the locations analyzed in *MLL-AF9* leukemias (*Figure 5E–F* and *Figure 5—figure supplement 2A–B*). Critically, we observed that binding of both TF at promoter regions dependent on Kat2a for H3K9ac was globally reduced in *Kat2a KO* leukemias (*Figure 5E–F*) as compared to WT

(Myc: 95% CI WT-KO enrichment 0.01273 to 1.589; 2-way ANOVA p<0.05 for genotype contribution; Gabpa: 95% CI WT-KO enrichment 0.2640 to 1.364; 2-way ANOVA p<0.01 for genotype contribution), thus suggesting that Kat2a may regulate binding of sequence-specific TF at the promoter regions it acetylates.

## Kat2a participates in AML maintenance through regulation of translation-associated genes

Having established that Kat2a can control the H3K9ac status, and putatively TF binding, of a subset of promoters, with impact on their bursting activity, we focused on the nature of the genes showing reduced frequency of bursting in *Kat2a KO* leukemias. We found that these were preferentially associated with translation categories (*Supplementary file 7*), including ribosomal proteins and translation initiation and elongation factors, some of which we showed to be depleted of TF binding upon Kat2a loss. Additionally, we could observe an enrichment of translation-associated gene expression signatures amongst undifferentiated cells relatively depleted in *Kat2a* KO leukemias (*Figure 6A*), suggesting that they may contribute to leukemia propagation and/or maintenance. We started by verifying that perturbed regulation by Kat2a had functional consequences for the translation machinery by performing polysomal profiling of *MLL-AF9*-carrying MOLM-13 AML cells in which KAT2A activity was inhibited by the MB-3 inhibitor (*Tzelepis et al., 2016*). Reassuringly, we observed a dramatic reduction in polysomal content (*Figure 6B*), indicating the functional impact of Kat2a transcriptional control, despite minimum changes in average gene expression. We then tested the effect on emergent protein synthesis in mouse MLL-AF9 leukemias by quantifying OP-Puro incorporation in *Kat2a* WT and KO phenotypic L-GMP. We saw qualitative and quantitative reductions in OP-Puro incorporation in *Kat2a* KO cells, with a bimodal distribution of low and high incorporating cells that was unique to *Kat2a*-depleted cells (*Figure 6C*), and significantly lower levels protein synthesis within the high OP-Puro distribution (*Figure 6D*). Altogether, the data suggest that phenotypic leukemia stem-like cells are less translationally active in the absence of *Kat2a*, which could explain their reduced functionality. We assessed functional impact in vitro by inhibiting the translation machinery of *Kat2a* WT and KO *MLL-AF9* transformed cells with the S6K1 inhibitor PF4708671 (*Pearce et al., 2010*). PF4708671 targets the TOR pathway and impedes translation initiation and elongation. Treatment of *MLL-AF9* transformed primary mouse BM cultured cells with PF4708671 in colony-forming assays recaptured the imbalance between compact and mixed colonies (*Figure 6E*) observed upon in vitro transformation of *Kat2a* KO cells. Taken together, the data indicate that regulation of the translational machinery at least partially mediates the imbalance between self-renewal and differentiation observed in *MLL-AF9*-driven leukemia upon *Kat2a* loss. It further supports the notion that fine control of transcriptional activity leading to changes in variability of transcript levels influences cell fate transitions and can be exploited by malignant cells for cancer maintenance.

## Discussion

In this study, we combine functional assays with single-cell transcriptional analysis and identify a requirement for the histone acetyl-transferase Kat2a in sustaining leukemia stem-like cells (LSC) and their downstream leukemia cellular structure in *MLL-AF9*-initiated AML. Loss of *Kat2a* increases cell-to-cell variability in transcription at all levels of mean gene expression, which is reflected in poor coordination in gene expression programs in *Kat2a* KO leukemia cells. Perturbation of transcription associates with an increased trend towards differentiation in vitro and loss of long-term functional LSC in vivo. However, the differentiation routes followed by *Kat2a* KO leukemic cells are aberrant and seem to follow multiple alternative dead ends that deviate from the seemingly linear pathway of *Kat2a* WT *MLL-AF9* cells. These characteristics are reminiscent of the perturbation of pluripotency we observed in Kat2a-inhibited mouse embryonic stem (mES) cells (*Moris et al., 2018*), where cells accumulate at the exit from pluripotency and only progress to differentiation with slow kinetics. Indeed, that may be the expected outcome of constitutively enhancing variability in gene expression, or transcriptional noise, which may need to be buffered downstream of a differentiation transition for differentiation to proceed (*Ahrends et al., 2014*).

Although it is surprising that we do not observe changes to normal hematopoietic stem and progenitor cells, the same was reported by others using a tissue-specific developmental (*Vav-Cre*) knockout (*Bararia et al., 2016*). Stem cell and developmental systems are typically robust to noise

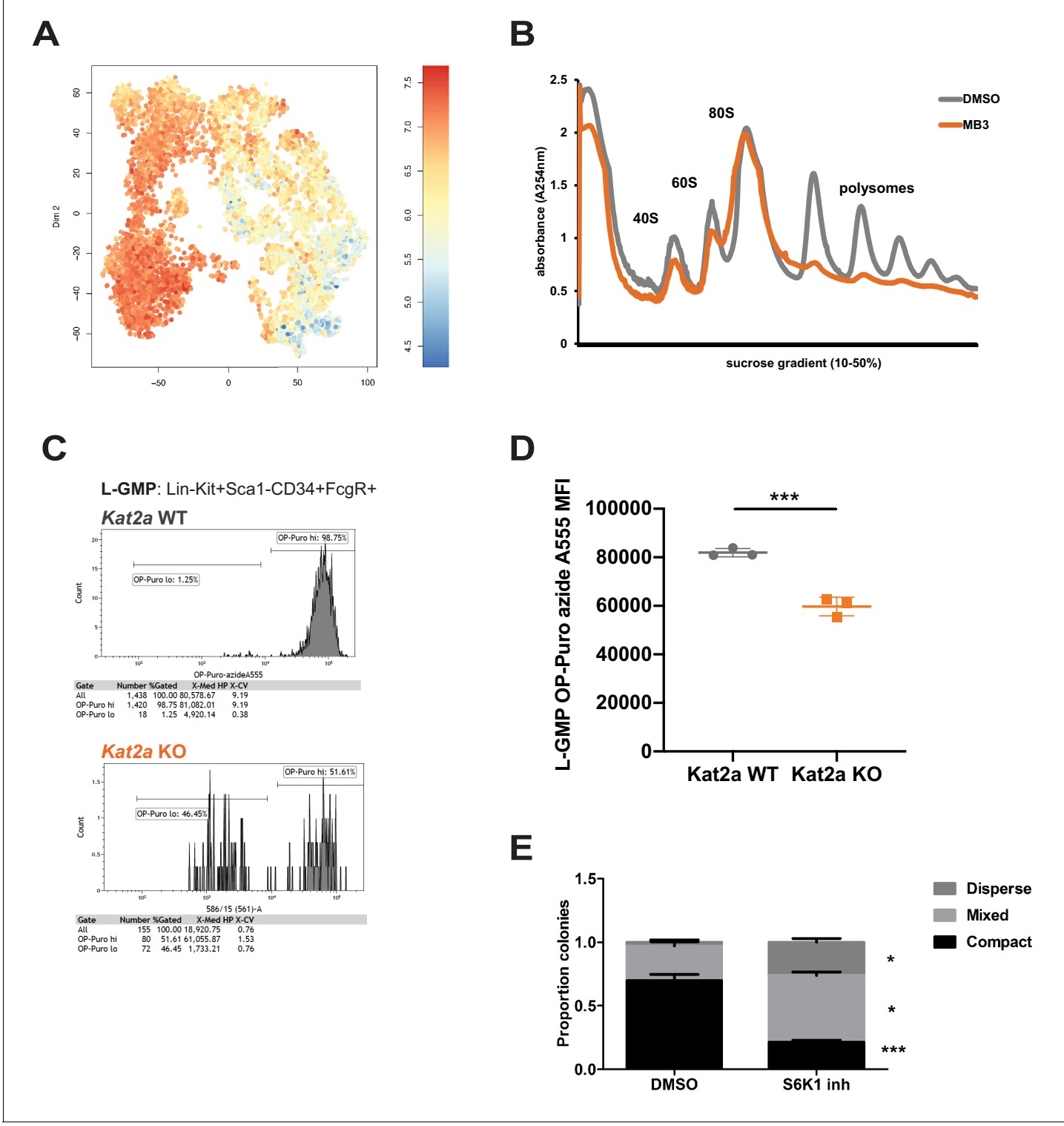

**Figure 6.** Kat2a regulates protein synthesis activity in MLL-AF9 leukemia stem-like cells. (**A**) Expression of translation-associated gene signatures in individual cells along the global *MLL-AF9* STEM-ID pseudotime trajectory. Trajectory representation as in *Figure 4A*, with both genotypes in the same plot. Gene signature defined as per the representation of gene sets MORF_EIF4E, MORF_EIF3S2, MORF_EIF4A2, MORF_EIF3S6 (MSigDB) in the Robust geneset. (**B**) Polysomal profiling of MOLM-13 cells upon overnight treatment with DMSO or the Kat2a inhibitor MB-3 (*Biel et al., 2004*) (200 μM); data are representative of 2 independent experiments. (**C**) Flow cytometry plot of OP-Puro incorporation by phenotypic L-GMP isolated from spleens of MLL-AF9 secondary leukemias WT or KO for the *Kat2a* gene. This pattern was observed in 2 out of 3 Kat2a KO leukemias analysed (0/3 WT). (**D**) Quantitation of protein synthesis rate in *Kat2a* WT and KO L-GMP as measured by OP-Puro incorporation. Mean ± SEM; n = 3 individual leukemia

*Figure 6 continued on next page*

*Figure 6 continued*

samples/genotype; *p<0.001. (E) Proportion of colonies types in CFC assays of *Kat2a* WT cells treated with DMSO vs. S6K1 PF4708671 inhibitor, mean + SEM, n = 3, ***p<0.001, *p<0.05. 2-tailed paired t-test performed in (D) and (E).

The online version of this article includes the following source data for figure 6:

**Source data 1.** Differential colony counts of MLL-AF9-transformed cells treated with PF4708671 S6K1 inhibitor.

perturbations (*Urban and Johnston, 2018*), and may have compensatory systems that buffer the loss of *Kat2a*. A putative compensatory mechanism would be the up-regulation of the *Kat2a* ortho-logue *Pcaf/Kat2b*, which is normally absent in HSC and progenitors. Whilst we did not observe up-regulation of *Kat2b* in leukemia cells, we did not specifically look for it in normal hematopoiesis, and cannot exclude that it may explain the difference between healthy and leukemic blood cells. We did note a very early reduction in HSC and their multilineage colony-forming potential upon *Kat2a* loss, but this was not sustained and did not affect engraftment potential, a more accurate measure of HSC activity. Also, the *Kat2a* KO hematopoietic system did not exhibit increased sensitivity to pro-longed treatment with the cytotoxic drug 5-fluorouracil (data not shown), suggesting that KO and WT HSC are similarly functional. Although we cannot exclude that serial or competitive transplanta-tion may reveal a defect in Kat2a KO HSC, our data are nevertheless supportive of a difference in the sensitivity of leukemia cells to *Kat2a* loss.

It has been proposed that cancer cells exist in a state of enhanced transcriptional activity that is required to sustain their oncogenic self-renewal programs (*Lin et al., 2012*). Amongst other factors, hyper-transcription has been associated with Myc, a known collaborator of Kat2a in transcriptional regulation (*Hirsch et al., 2015*), which is broadly required in AML (*Delgado and León, 2010*), includ-ing in *MLL* fusion-driven disease (*Schreiner et al., 2001*). Accordingly, we found Myc to be depleted in a subset of promoters targeted by Kat2a. However, transcription factor depletion at promoters was not exclusive to Myc, and we made similar and indeed more extensive observations with another broad metabolic regulator, Gabpa, which has been previously characterized as a regulator of CML self-renewal (*Yang et al., 2013*; *Yu et al., 2012*) and associated with Kat2a-containing com-plexes (*Krebs et al., 2011*), but for which a requirement in AML has not been established. Of inter-est, loss of both transcription factors analysed have severe consequences to normal hematopoiesis, which are of HSC depletion in the case of Gabpa (*Manukjan et al., 2016*; *Yu et al., 2011*) and impaired differentiation with accumulation of defective phenotypic HSC in the case of Myc (*Wilson et al., 2004*). Notably, none of these roles is phenocopied by *Kat2a* loss. We thus suggest that Kat2a acts through co-option of the transcriptional machinery present at target loci, rather than rely on a unique conserved transcription factor circuit, to exert its pleiotropic activating effects in leukemia cells. This said, we did not observe a monotonous reduction in transcription factor binding across all loci analysed, suggesting that the presence of Kat2a may facilitate transcription factor recruitment and/or binding in a probabilistic manner, a view compatible with a role for Kat2a in sta-bilizing rather than initiating transcription (*Jin et al., 2014*). Live-cell imaging of transcription factor recruitment to individual loci (*Donovan et al., 2019*) and/or single-cell ChIP (*Ai et al., 2019*; *Hainer et al., 2019*; *Ku et al., 2019*), currently undergoing significant development, will be central to definitively test this hypothesis. An alternative, albeit not mutually exclusive, explanation is that the role of Kat2a in transcription factor binding or recruitment is influenced by the chromatin context in which it acts, including the post-translational modifications it catalyses as well as the modifications introduced by other factors.

Kat2a catalyzes acetylation of Lys nine in Histone 3 (H3K9ac), a chromatin mark associated with maintenance, but not initiation, of locus transcription. Kat2a is able to catalyze acetylation of addi-tional lysine residues in cell-free systems (*Kuo and Andrews, 2013*), but its loss in vivo more specifi-cally affects H3K9ac, particularly so in the vicinity of transcriptional start sites (*Wang et al., 2018*). Compatible with these observations, Kat2a loss in the context of MLL-AF9 leukemia impacted H3K9ac specifically at gene promoters. In addition, H3K9ac was specifically lost at gene loci that do not exhibit additional activating acetylation marks such as H3K27ac. Whilst the specific meaning of single vs. double-acetylated regions is unclear, one possibility is that the presence of H3K27ac marks genomic regions that also function as enhancers of more distant genes, suggesting that Kat2a may be strictly required at promoters. In agreement with this view, we did not observe loss of H3K9ac at

H3K4me1-positive enhancers. Instead, we observed a gain in H3K27ac at both H3K4me3 promoters and H3K4me1 enhancers as a single acetylation mark, which may help explain the minimal consequences of *Kat2a* loss in terms of average gene expression, and highlight the specific role of Kat2a-dependent H3K9ac of promoters in stabilizing transcriptional activity. Gain of H3K27ac may reflect differentially regulated or compensatory acetylation by other histone acetyl-transferases in the absence of Kat2a. In light of the promoter vs. enhancer specificity of the changes observed, it will be interesting to investigate to what extent reprogramming of acetylation marks reflects proximal reconfiguration of enhancer-promoter interaction via Ctcf binding. Ctcf loss has been shown to increase gene expression CV with moderate or no differences in mean expression (*Ren et al., 2017*), a pattern akin to our observations upon *Kat2a* loss. Whereas their study specifically investigated sequence-driven loss of Ctcf binding at proximal, intra-TAD enhancer regions, we observed that H3K9ac-depleted promoters in *Kat2a* KO leukemia cells had a significant association with experimental Ctcf binding in ENCODE experiments, and we speculate that Ctcf may be dislodged to enhancers and promote asymmetric distribution of histone acetylation marks, with dysregulation of locus control. Of note, too, is the fact that despite the almost complete knockout of *Kat2a* expression, the loss of H3K9ac, although specific in terms of chromatin context, is far from dramatic. This is similar to recent observations in embryoid bodies (*Wang et al., 2018*) and suggests that Kat2a requirement for H3K9ac is not absolute, although it may be locus-specific. Other acetyltransferases may, either normally or compensatorily, contribute to H3K9 acetylation in at least some locations, and it will be interesting to understand the parameters that determine specific dependency on Kat2a activity and its unique consequences to transcription.

Variability in gene expression levels reflects regulation of locus activity, and whilst specific contribution from enhancers has been proposed (*Fukaya et al., 2016*) and remains an area of active investigation (*Larsson et al., 2019*) the role of promoter configuration and sequence has been more extensively characterized in multiple model systems (*Antolović et al., 2017*; *Faure et al., 2017*; *Zoller et al., 2015*). In most if not all *loci*, transcriptional activity is discontinuous, with promoters cycling between active (ON) and inactive (OFF) states. Self-limited bursts of transcriptional activity are characterized by the burst frequency, reflecting the rate of OFF-to-ON transitions, and the burst size, which captures the number of mRNA molecules produced during each burst. In yeast, regulation of both burst parameters is dependent on H3K9ac at specific gene locations: gene body acetylation regulates burst size; promoter H3K9ac associates primarily with burst frequency (*Weinberger et al., 2012*). Furthermore, in yeast, promoter H3K9ac is deposited by the Kat2a orthologue and founder histone acetyl-transferase Gcn5, and removed by the Sin3a orthologue Rpd3(L) deacetylase complex. Loss of Gcn5 decreases burst frequency across multiple yeast loci and has been modelled to increase intrinsic transcriptional noise, a finding we capture in mammalian cells in the present study. Whilst our study specifically links promoter H3K9ac to regulation of burst frequency in mammalian cells, recent work by the Naëf lab has shown that locus-specific manipulation of promoter, but not distal or enhancer, H3K27ac can also change transcriptional bursting frequency (*Nicolas et al., 2018*). Although the Naëf study has not specifically manipulated H3K9ac levels, it did reveal an association between promoter H3K9ac and frequency of locus activation, which agrees with our own observations. Whether other residue-specific acetylations of promoters (or indeed enhancers) can produce the same effect remains to be determined, and this knowledge will undoubtedly deepen current understanding of transcriptional regulation. Moreover, Kat2a was recently shown to catalyze other acyl-modifications of lysine residues, namely succinylation (*Wang et al., 2017*), which also associates with transcriptional activation (*Tong et al., 2020*). Characterization of the exact mechanistic consequences of additional acylations and their interaction with the better characterized lysine acetylations is still lacking, not least due to lack of modification and residue-specific antibodies for the newly-identified marks. It is possible that their loss also contributes to the changes in transcriptional regulation seen upon *Kat2a* KO, and could for example explain why the reduction in frequency of bursting, although more strongly associated with loss of H3K9ac, is not exclusive to sites depleted of this modification. Additionally, it remains possible that loss of Kat2a may impact other residue-specific acylations more dramatically than its specific effect on H3K9ac and the effects of their combined loss more completely link all the effects observed. Our lab has recently developed a KAT2A-dCas9 fusion capable of catalyzing targeted acetylation events (data not shown), which will be instrumental in the mechanistic understanding of individual acetylation events and specific sequences in regulating bursting activity. It should also allow us to probe

other candidate acyl modifications to unveil their unique and combined effects on transcriptional bursting.

Somewhat unexpectedly, we found that the genes regulated by Kat2a at the level of promoter acetylation, and which responded to *Kat2a* loss with decreased frequency of bursting, were specifically associated with ribosomal assembly and translation activity. Similar categories have been shown to be regulated by non-catalytic components of the Kat2a SAGA complex in controlling ESC pluripotency (*Seruggia et al., 2019*), reinforcing the notion that Kat2a complexes impact general metabolic processes in multiple cell types, and that these general processes can specifically influence cell fate transitions. Importantly, we demonstrated that *Kat2a*-depleted leukemia stem-like cells (phenotypic L-GMP) have reduced protein synthesis activity, putatively due to a perturbation of polysomal assembly consequent to variability in levels of ribosomal proteins. Moreover, perturbation of the translational machinery could re-capture the enhanced in vitro differentiation of leukemia cells observed upon *Kat2a* depletion, suggesting that alterations in protein synthesis activity may indeed be central to exit from leukemia self-renewal. In agreement, Morrison and collaborators (*Signer et al., 2014*) had previously reported that impaired protein synthesis upon genetic depletion of the ribosomal protein machinery impedes leukemia self-renewal, whilst having non-linear dose-dependent effects on normal hematopoiesis, mimicking our own observations in the *Kat2a* KO setting.

Future studies directing Kat2a catalytic activity to single or multiple loci will illuminate individual *vs.* global target gene contributions to the leukemia phenotype. However, it is tempting to speculate that the generic nature of the programs impacted by Kat2a at the level of transcriptional noise may configure an underlying propensity towards execution of cell fate transitions, which can be of a different nature in different biological contexts. Analysis of the impact of Kat2a target programs in other malignant and normal stem cell systems, or at different stages of leukemia progression will test this hypothesis. It will also be interesting to determine if other candidate regulators of transcriptional noise produce similar effects and can be exploited therapeutically in AML, as well as in other hematological and non-hematological malignancies.

## Materials and methods

### Generation and analysis of *Kat2a* conditional knockout mice

*Kat2a*$^{Fl/Fl}$ conditional knockout mice (*Lin et al., 2008*) (MGI:3801321) were bred with *Mx1-Cre* $^{+/-}$ transgenic mice (*Kühn et al., 1995*), in a C57Bl/6 background. Littermates were genotyped for *Kat2a* LoxP sites (forward: CACAGAGCTTCTTGGAGACC; reverse: GGCTTGATTCCTGTACCTCC) and for *Mx1-Cre*: (forward: CGTACTGACGGTGGGAGAAT; reverse: TGCATGATCTCCGGTATTGA): Ear notch biopsies were digested using KAPA express extract (Sigma Aldrich) and KAPA2G ROBUST HS RM Master Mix (2x) (Sigma Aldrich). PCR cycling protocol: 95C, 3 min; 40x (95°C, 15 s; 60°C, 15 s; 72°C, 60 s); 72°C, 60 s. DNA products were run on a 1% Agarose Gel in TAE (1x), at 100V and visualized using an AlphaImager UV transilluminator (Protein Simple). Cre-mediated recombination was induced in 6–10 week-old mice by administration of 5 alternate-day intraperitoneal injections of poly (I)- poly(C) (pIpC), 300 μg/dose. After pIpC treatment, animals were identified as *Kat2a* WT = *Kat2 a*$^{Fl/Fl}$ * *Mx1-Cre* $^{-/-}$ and *Kat2a* KO = *Kat2 a*$^{Fl/Fl}$ * *Mx1-Cre* $^{+/-}$. Excision efficiency was determined by qPCR of genomic DNA (gDNA) from Peripheral Blood (PB), Spleen (Sp) or Bone Marrow (BM). gDNA was extracted using Blood and Tissue DNA easy Kit (Qiagen) and quantified by Nanodrop (Thermo Scientific). qPCR analysis used Sybr Green Master Mix (Applied Biosystems) and two sets of primers (*Figure 1A*): *Kat2a*-IN11 (forward: CAACTTCCCCAAGGTATGGA; reverse: CGGGGACC TTAGACTTGTGA), within the excised region; *Kat2a*-OUT18 (forward: AGTCTGGGCTGTTTCCATGT; reverse: GCCCGTTGTAGAATGTCTGG), distal to the second LoxP site. Expression levels were determined by the Pfaffl method following normalization to *Kat2a*-OUT. PB was collected by saphenous vein and differential blood cells counts were determined using a Vet abc automated counter (Scil Animal Care, Viernheim, Germany). Mice were kept in an SPF animal facility, and all experimental work was carried out under UK Home Office regulations. Animal research was regulated under the Animals (Scientific Procedures) Act 1986 Amendment Regulations 2012 following ethical review by the University of Cambridge Animal Welfare and Ethical Review Body (AWERB).

## Isolation of mouse BM stem and progenitor cells

BM was isolated from mouse long bones as described before (Pina et al., 2015). Following red blood cell lysis, total BM suspension was depleted of differentiated cells using a cocktail of biotinylated lineage (Lin) antibodies (Table 1) and streptavidin-labeled magnetic nanobeads (Biolegend), according to manufacturers' instructions. Cells were directly used in transplants, colony-forming assays or flow cytometry for analysis of normal hematopoiesis. For leukemia studies, cells were cultured overnight at 37°C 5% $CO_2$ in RPMI supplemented with 20% Hi-FBS (R20), 2 mg/mL L-Glutamine, 1% PSA, 10 ng/mL of murine Interleukin 3 (mIL3), 10 ng/mL of murine Interleukin 6 (mIL6), and 20 ng/mL of murine Stem Cell Factor (mSCF) (cytokines from Peprotech) (supplemented R20), followed by retroviral transduction.

## Colony forming cell (CFC) assays

For analysis of normal progenitors, sorted mouse BM cells were plated at a density of 200–400 cells/ plate in duplicates, in MethoCult GF M3434 (STEMCELL Technologies). Colonies were scored at 7–9 days. For analysis of MLL-AF9 leukemia, retroviral-transduced BM cells were plated in M3434 at an initial density of 10000 cells/condition and scored and re-plated every 6–7 days. Re-plating was performed up to passage 9, with 4000 cells/condition used from plate 3. CFC assays from mouse MLL-AF9 transformed lines were seeded in M3434 and scored 6–7 days later. RPS6K inhibition studies were set by adding 3.3 μL DMSO, either as vehicle or with a final concentration of 3.5 μM of PF4708671 (Tocris), directly to the methylcellulose medium, with mixing prior to cell addition.

**Table 1.** Antibodies used in flow cytometry analysis and cell sorting.

| Antibody | Fluorochrome | Catalogue # | Clone | Dilution | Supplier |
|---|---|---|---|---|---|
| CD45.1 | FITC | 110705 | A20 | 1:200 | BioLegend |
| CD45.2 | AF700 | 56-0454-81 | 104 | 1.:200 | Ebioscience |
| CD45R/B220 (Lin)* | Biotin | 103204 | RA3-6B2 | 1:300 | BioLegend |
| Ter119 (Lin)* | Biotin | 116204 | Ter119 | 1:300 | BioLegend |
| Gr1 (Lin)* | Biotin | 108404 | RB6-8C5 | 1:300 | Biolegend |
| CD3e (Lin)* | Biotin | 100304 | 145–2 C11 | 1:300 | BioLegend |
| CD11b (Lin)* | Biotin | 101204 | M1/70 | 1:300 | BioLegend |
| CD11b/Mac1 | AF700 | 101222 | M1/70 | 1:200 | BioLegend |
| CD11b/Mac1* | PE-Cy7 | 25-0112-81 | M1/70 | 1:200 | Ebioscience |
| CD16/32/FcγR* | PE | 101308 | 93 | 1:100 | BioLegend |
| CD16/32/FcγR | PerCP-Cy5.5 | 101323 | 93 | 1:100 | BioLegend |
| CD34* | APC | 128612 | HM34 | 1:100 | BioLegend |
| CD34 | AF700 | 560518 | RAM34 | 1:100 | BD |
| CD117/c-Kit* | APC-Cy7 | 105826 | 2B8 | 1:50 | BioLegend |
| CD117/c-Kit | APC-eF780 | 47-1171-82 | 2B8 | 1:100 | Ebioscience |
| Sca1* | PE-Cy7 | 108114 | D7 | 1:100 | BioLegend |
| Gr1 | AF700 | 108422 | RB6-8C5 | 1:200 | BioLegend |
| Gr1 | PB | 108430 | RB6-8C5 | 1:100 | BioLegend |
| CD135/Flt3* | PE | 135305 | A2F10 | 1:100 | BioLegend |
| CD105* | PE | 562759 | MJ7/18 | 1:100 | BD |
| CD150* | Af647 | 562647 | Q38-480 | 1:100 | BD |
| CD41* | Biotin | 13-0411-81 | eBioMWReg30 | 1:100 | Ebioscience |
| Streptavidin* | BV421 | 405226 | | 1:200 | BioLegend |
| Streptavidin | BV510 | 405233 | | 1:200 | BioLegend |

## In vivo analysis of leukemia initiation and engraftment

For analysis of normal hematopoiesis, $10^6$ *Kat2a* WT or *Kat2a* KO cKit+ cells were intravenously injected via tail vein into lethally irradiated (2*5.5Gy) CD45.1 recipient mice. At the described time-points, BM and Sp were collected and processed into a single-cell suspension for surface marker staining and flow cytometry analysis. For AML induction, we transplanted $1.5 \times 10^6$ cKit+ *Kat2a* WT or *Kat2a* KO cells transduced with *MSCV-MLL-AF9-IRES-YFP*, intra-venous into lethally irradiated (2* 5.5Gy) CD45.1 recipient mice. The number of recipients used was determined by the numbers of cells available post-retroviral transduction and the transduction efficiency estimated by flow cytometry at the point of injection, aiming at a minimum of $1 \times 10^5$ YFP+ cells/recipient and the same number of YFP+ cells delivered to all recipients. The investigators were blinded as to the group allocation, with the injections performed by an investigator not involved in sample preparation, or in subsequent animal follow-up and tissue collection. Upon signs of illness and following human end-point criteria, animals were culled, tissue samples collected for histology analysis, and BM and Sp processed into single-cell suspensions. Flow Cytometry analysis and DNA extraction were performed. Data collection was performed using general identification numbers with no reference to the experimental group. For limiting-dilution analysis, $5*10^2$ - $5 \times 10^4$ cells from primary leukemia pooled BM of each genotype were transplanted into sub-lethally irradiated (1*5.5Gy) CD45.1 recipient mice (3–4/dose and genotype). Numbers of animals used were contingent on availability on CD45.1 recipients aiming at no less than 10 recipients per genotype divided between 3 cell doses to allow for limiting dilution statistical analysis.

## Retroviral transduction

Retroviral construct *MSCV-MLL-AF9-IRES-YFP* was previously described (*Fong et al., 2015*). For viral particle production, Human Embryonic Kidney (HEK) 293 T cells were seeded at $2.5 \times 10^6$ cells/10 cm dish in DMEM supplemented with 10% Hi-FBS, 2 mg/mL L-Glutamine, 1% PSA and cultured overnight at 37℃ 5% $CO_2$. The following day, a transfection mix [per plate: 47.5 µL of TranSIT (Miros), 5 µg of packaging plasmid psi Eco vector (5 µg), retroviral vector (5 µg) and 600 µL of Opti-mem Medium (Gibco)] was prepared according to manufacturer's instructions and added dropwise to cells followed by plate swirling and overnight culture at 37℃ 5% $CO_2$. Medium was replaced with R20 the next day. At 24 and 48 hr after R20 replacement, medium was collected and filtered through a 0.45 µM syringe filter, and viral particle suspension medium was added to BM cells. BM cells from 6 to 10 week old *Kat2a* WT and *Kat2a* KO mice were collected and Lineage-depleted as described above (Isolation of mouse BM stem and progenitor cells), and cultured overnight at 37℃ 5% $CO_2$ in supplemented R20. For viral transduction, BM cells were briefly centrifuged at 400G, 5 min, and viral particle suspension medium supplemented with 10 ng/mL mIL3, 10 ng/mL mIL6, and 20 ng/mL mSCF added to a final density of $10^6$ cells/mL. Cells were plated in 6-multiwell plates and centrifuged for 1 hr at 2000 rpm, 32℃. After, cells were incubated for 4 hr at 37℃ 5% $CO_2$. A second round of viral transduction was performed, with post-centrifugation incubation performed overnight. Next day, cells were collected, pelleted and washed three times with PBS (2x) and R20 (1x). YFP level was accessed by Flow Cytometry in a Gallios Analyser (Beckman Coulter).

## Establishment of MLL-AF9 transformed cell lines

MLL-AF9 clonal liquid cultures were set up using *MLL-AF9* retrovirus-transduced primary BM cells (see Retroviral Transduction section). Transformed cells enriched in vitro by 3 rounds of serial plating (CFC assays) were maintained in R20 supplemented on alternate days with mSCF, mIL3 and mIL6, all at 20 ng/mL. Cells were cultured at $2*10^5$ cells/ml and passaged when they reached a density of $1*10^6$/ml.

## Flow cytometry

Cell surface analysis of BM and Sp was performed using a panel of antibodies marked with * described in *Table 1*, as per the following sorting strategies: HSC - Lin⁻ cKit⁺ Sca1⁺ CD34⁻ Flt3⁻; MPP: Lin⁻ cKit⁺ Sca1⁺ CD34⁺ Flt3⁻; LMPP: Lin⁻ cKit⁺ Sca1⁺ CD34⁺ Flt3⁺; CMP: Lin⁻ cKit⁺ Sca1⁻ CD34⁺/low CD16/32low; GMP: Lin⁻ cKit⁺ Sca1⁺ CD34⁺ CD16/32high; MEP: Lin⁻ cKit⁺ Sca1⁺ CD34⁻ CD16/32⁻; Lin⁻: CD3e⁻ B220⁻ Gr1⁻ CD11b⁻ Ter119⁻. Data were acquired on Gallios (Beckman Coulter)

or LSRFortessa (BD) cytometers; data analysis used Kaluza software (Beckman Coulter). Cell sorting was performed on Influx or AriaII instruments (both from BD).

## Measurement of protein synthesis

Six million cells each from the spleens of 3 individual secondary leukemia samples per genotype were cultured overnight in supplemented R20 to allow recovery after thaw. O-propargyl-puromycin (OP-Puro, Thermo Fisher Scientific) was added directly to 80% of each culture at a final concentration of 50 μM and incubated for 1 hr at the end of the culture period; the remainder was treated with PBS and processed in parallel as a control. After incubation, cells were washed twice in ice-cold PBS without $Ca^{2+}$ or $Mg^{2+}$ (Sigma), and resuspended in PBS/10%FBS for cell surface staining with Lineage markers (Gr1/Mac1-AF700), c-Kit-APC-ef780, Sca1-PE-Cy7, CD34-APC, CD16/32-PerCP-Cy5.5 (see *Table 1*) for 30 min on ice. After washing, cells were fixed in 1% paraformaldehyde (PFA) in PBS for 15 min on ice, protected from light, washed, and then permeabilized in PBS/3% FBS/0.1% saponin (permeabilization buffer) at room temperature, in the dark, for 5 min. Cells were washed and used immediately in the azide-alkyne cyclo-addition reaction with Click-iT Cell Reaction Buffer Kit (Thermo Fisher Scientific; C10269) and Alexa Fluor 555-Azide (Thermo Fisher Scientific; A20012) with a master reaction solution freshly prepared as per manufacturer's instructions for immediate use. Alexa Fluor 555-Azide was used at a final concentration of 5 μM. The reaction proceeded in the dark at room temperature for 30 min; cells were washed twice in permeabilization buffer and then counterstained with DAPI 3.3 μg/ml in PBS for 5 min prior to flow cytometry analysis. We did not observe any effect of cell cycle status on differential OP-Puro labeling.

## Polysomal profiling

MOLM-13 cells (ID: CVCL_2119) were grown to an approximate density of $1 \times 10^6$ cells/mL, treated with cycloheximide (100 μg/mL) for 15 min, washed in ice-cold PBS and stored at −80°C. Cells were lysed in buffer A (20 mM HEPES pH 7.5, 50 mM KCl, 10 mM $(CH_3COO)2$ Mg, EDTA-free protease inhibitors (Roche), supplemented with cycloheximide 100 μg/mL, 1 mM PMSF, 100 U/mL RNase inhibitor (Promega), 1% (vol/vol) sodium deoxycholate, and 0.4% (vol/vol) NP-40) at $10^8$ cells/mL for 10 min on ice. Lysates were cleared by centrifugation (8000 g for 5 min at 4°C) and 3 A254nm units loaded onto a 10–50% (wt/vol) sucrose gradient in buffer A in Polyallomer 14 × 95 mm centrifuge tubes (Beckman). After centrifugation (Beckman SW40Ti rotor) at 260 900 g for 3 hr at 4°C, gradients were fractionated at 4°C using a Gilson Minipulse three peristaltic pump with continuous monitoring (A254nm). Samples were analysed using a Brandel gradient fractionator, the polysome profiles were detected using a UV monitor (UV-1, Pharmacia) at A254, and 0.5 mL fractions were collected. The electronic outputs of the UV-1 monitor and fraction collector were fed into a Labjack U3-LV data acquisition device with an LJTick-InAmp preamplifier.

## Quantitative real time PCR (Q-RT-PCR)

Total RNA was extracted using Trizol Reagent (Invitrogen). RNA from equal numbers of cells was reverse-transcribed using Superscript II (Invitrogen), following manufactures' instructions. Complementary (c)DNA was analyzed in duplicate or triplicate by qPCR using Taqman gene expression assays (*Ppia*; Mm03024003_g1; *Hprt*: Mm01545399_m1; *Kat2a*: Mm00517402_m1) and Taqman Gene Expression Mastermix (Applied Biosystems). Gene expression levels were determined by the Pfaffl method following normalization to Reference gene, as stated. For exon 2–18 in-frame products, qPCR using Sybr Green Master Mix (Applied Biosystems) was performed in triplicates. Primers used were: *Kat2a* Exon 1–2 (forward: GTCTTCTCAGCTTGCAAGGCC, reverse: AAAGGGTGCTCA-CAGCTACG); *Kat2a* Exon2-18 (forward: GTAGCTGTGAGCACCCTTTGG, reverse: TTCGCTGTC TGGGGGATTGT); *Kat2a* Exon18 (forward: CTCATCGACAAGTAGCCCCC; reverse: GTCCCTGGC TGGAGTTTCTC).

## Single-cell RT-PCR

*MLL-AF9* secondary leukemia cells from *Kat2a* WT and KO backgrounds were freshly thawed, stained with Lin cocktail, c-Kit-APC-ef780, Sca1-PE-Cy7, CD16/32-PerCP-Cy5.5, CD34-AF700 and Streptavidin BV510 (see *Table 1*), plus dead cell exclusion with Hoechst 32558 (Thermo Fisher Scientific), sorted as Lin⁻c-Kit⁺Sca1⁻CD16/32⁺ and single-cell deposited into 96-well plates containing 3 μl/

well of 0.67% NP-40 and 2U of RNasin Plus (Promega) in RNase-free water. Plates were vortexed after deposition, centrifuged for lysate collection, and frozen immediately in dry ice. Plates were stored at −80°C. Upon thaw, cDNA was synthesized from the single-cell lysates using 5U Superscript II (Invitrogen) and 1 µM each of gene-specific primers for *Hprt* and *Kat2a* (outer reverse, *Table 2*) in a 10 µl reaction mix as per the manufacturer's protocol (42°C, 1 hr; 70°C inactivation, 15 min). The total cDNA reaction was used in a 50 µl first round PCR with duplexed outer forward primers for *Hprt* and *Kat2a* at a final concentration of 200 nM and 1.25U HotStar Taq (Qiagen) in a reaction mix as per manufacturer's protocol. Cycling conditions: 95°C, 15 min; 40*(94°C, 1 min; 60°C, 1 min; 72°C, 2 min); 72°C, 5 min; 25°C, 30 s. Two µl of the first-round product were amplified in each of two separate second round PCR using nested primers for the two individual genes (*Table 2*). Reaction mixes as per the first round PCR, but in a final volume of 25 µl. Cycling conditions: 95°C, 15 min; 40*(94°C, 30 s; 60°C, 1 min; 72°C, 1 min); 72°C, 5 min; 25°C, 30 s. Second-round products were run in a 2% agarose TAE1x gel at 50V, 1 hr, stained for 45–60 min in SYBRSafe solution (Thermo Fisher Scientific) in double-distilled water, and DNA visualized on a BioRad Imager.

## Chromatin immunoprecipitation (ChIP)

Pools of total BM cells from MLL-AF9 *Kat2a* WT and *Kat2a* KO primary leukemia samples (ChIP-sequencing) and of total BM or spleen cells from MLL-AF9 *Kat2a* WT and *Kat2a* KO secondary leukemias (ChIP-qPCR) were crosslinked with 1% Formaldehyde Solution (Sigma Aldrich) for 10 min at room temperature (RT), with gentle rotation (50 rpm). Fixation was stopped with Glycine, and cells incubated for 5 min, RT, with gentle rotation (50 rpm), followed by two washing steps in ice-cold PBS. Cell pellets were resuspended in Lysis buffer (20 mM Hepes pH 7.6, 1% SDS and 1/100 Protease Inhibitors cocktail (PIC, Sigma Aldrich) followed by Nuclei preparation. Chromatin pellets were sheared in a Bioruptor Pico Plus (Diagenode) in TPX tubes, using 3 runs of 11 cycles (Cycle: 30 s ON 30 s OFF) on high setting. A short spin was performed between runs and samples were transferred to new TPX tubes. 2–10% of total sheared chromatin was kept for input reference. Immunoprecipitation was set up using Dilution Buffer (0.15% SDS, 1% Triton X-100, 1.2 mM EDTA, 16.7 mM Tris pH8 and 167 mM NaCl), PIC, and the respective antibody (*Table 3*) and the sheared chromatin incubated overnight at 4°C with rotation. On the following day, protein A/G magnetic beads were pre-cleared with Dilution Buffer supplemented with 0.15% SDS and 0.1%BSA, then mixed with immunoprecipitation mix and incubated for at least 4 hr at 4°C with rotation. Chromatin-Antibody-Beads mixes were sequentially washed with ChIP Wash1 (2 mM EDTA, 20 mM Tris pH8, 1% Triton X-100, 0.1% SDS and 150 mM NaCl), ChIP Wash2 (2 mM EDTA, 20 mM Tris pH8, 1% Triton X-100, 0.1% SDS and 500 mM NaCl), ChIP Wash3 (1 mM EDTA and 10 mM Tris pH8) and captured on a magnetic rack. Captured beads were incubated for 20 min with rotation in freshly prepared Elution Buffer (1% SDS and 0.1M NaHCO$_3$). Supernatants were collected and decrosslinking performed overnight. DNA was column-purified using DNA clean and concentrator TM 5 KIT (Zymo Research), according to manufacturer's instructions, using 20 µL Zymo Elution Buffer. DNA quality control was performed using DNA Qubit 2.0/3.0 (Invitrogen); DNA fragment size <500 bp (typically 200–300 bp) was confirmed by gel electrophoresis of input material on a 1.5% agarose/TAE gel.

For ChIP-qPCR, eluted DNA samples were diluted 3–5 times and tested by SYBR green qPCR (primer sequences in *Table 4*) using 2 µl diluted DNA per triplicate reaction. Peak enrichments were quantified by the $-2^{\Delta\Delta Ct}$ method relative to mouse IgG (Gabpa ChIP) or rabbit IgG (Myc ChIP), using the intergenic region in mouse chromosome 1 (*mChr1*) as a reference. Duplicate to quadruplicate ChIP experiments were analyzed. For ChIP-sequencing analysis of histone modifications, DNA from duplicate immunoprecipitation experiments and the respective input material underwent library

**Table 2.** Primers used for single-cell RT-PCR analysis.

|  | Forward | Reverse |
| --- | --- | --- |
| *Hprt* (outer) | GGGGGCTATAAGTTCTTTGC | TCCAACACTTCGAGAGGTCC |
| *Hprt* (nested) | GTTCTTTGCTGACCTGCTGG | TGGGGCTGTACTGCTTAACC |
| *Kat2a* (outer) | CTTCCGCATGTTTCCCACTC | GATGTCTTTGGAGAAGCCCTG |
| *Kat2a* (nested) | GGAAATCGTCTTCTGTGCCG | TTTGGAGAAGCCCTGCTTTTTG |

**Table 3.** Antibodies used in chromatin immunoprecipitation (ChIP).

| Antibody | Catalogue# | Supplier |
| --- | --- | --- |
| H3K27ac | Ab4729 | Abcam |
| H3K9ac | 07–352 | Millipore |
| H3K4me3 | Ab8580 | Abcam |
| H3K4me1 | Ab8895 | Abcam |
| Myc | sc-764 x | Santa Cruz Biotechnology |
| Gabpa | sc-28312 | Santa Cruz Biotechnology |
| rabbit IgG | 12–370 | Millipore |
| mouse IgG | sc-2025 | Santa Cruz Biotechnology |

preparation with the NextFlex Rapid DNA kit (12 cycles) at the MRC/WT Cambridge Stem Cell Institute Genomics Core Facility. After quality control, libraries were sequenced on an Illumina HiSeq 4000 sequencer at the CRUK Cambridge Research Institute, using 50bp-single-end sequencing.

## ChIP-seq data analysis

Raw ChIPseq reads were analyzed on the Cancer Genomics Cloud (CGC) platform (*Lau et al., 2017*). Reads were aligned to the mm10 mouse genome using the Burrows-Wheeler Aligner (BWA). Peaks from aligned reads were obtained using MACS2 peak calling algorithm with a significance q-value of 0.05. The deepTools bamCoverage command (*Ramírez et al., 2016*) was used to determine reads enrichment relative to input; only ChIP-seq samples with clear separation from control were retained, with exclusion of one H3K4me1 and one H3K27ac replicate; consensus peaks were used for duplicate samples. To analyze changes in acetylation patterns at promoters and enhancers, H3K4me3 and H3K4me1 peaks respectively, were crossed with H3K9ac-only, H3K27ac-only and dual acetylation peaks from the corresponding genotypes. H3K9ac-only peaks associated with me3 were used for further analysis. Genomic peaks were obtained for *Kat2a* WT and *Kat2a* KO genotypes separately using Bedtools intersect (*Quinlan and Hall, 2010*) and H3K4me3 K9ac peaks exclusive to WT retained as putative Kat2a peaks. Peak locations were converted to fastq sequences using UCSC table browser tool (*Karolchik, 2004*). Genomic Regions Enrichment of Annotations Tool, GREAT (*McLean et al., 2010*) was used to assign gene identities to the fastq sequences associated with putative Kat2a peaks, with gene promoter peaks called within - 1 kb to + 500 bp of the transcription

**Table 4.** Primers used for ChIP-quantitative PCR analysis.

| | Forward | Reverse |
| --- | --- | --- |
| *Arpc3* | GTGCGTTTATTCCTTCCGCC | TCGAATGCTTACCGGCATCT |
| *Cct4* | TGGTCTCGTTTGCAGCTTTC | TGCAGGAGACGAACTAAGGA |
| *Cct7* | AACGCTCACATCCTCCGTT | GTAGGCACAACCTGACAACC |
| *Clptm1l* | GTAGGCACAACCTGACAACC | GGGTAACAAGAGAGCAGCAGA |
| *Eif4e* | GCAGACCACATCAACGACTCT | TCTTTTCGCCTCCCACCATT |
| *Gadd45g* | GGCATCGACTCTGACCTTGT | CGCTATGTCGCCCTCATCTT |
| *Kdelr2* | CCTTGAGTGTGGCCGTCTAA | TCAATGGTGACGTGGAGCAA |
| *mChr1 (intergenic)* | CATAGATGAAGCTGCCACATAGGT | GTGGGCAAGGACAAAGCATTA |
| *Parl* | CTTCGCCAGGCTCAATCTCA | ATACACACCAAGGGGCCTGA |
| *Pcbp1* | AGAGCGCCTTGTGCTTTCTT | CTGGTCCTTTCGGCCAAGTA |
| *Ralbp1* | GTGTTGACTTGCGGGAAACT | GCGGCTTTAACTCGGGTATG |
| *Rps15* | TCCGCAAGTTCACCTACCGT | TCTGGCTCTATTTCCAGCACC |
| *Rps7* | GCTTAGAAAGAGGGACGGCT | CGGTTTCCACCCACCTACTT |
| *Zbtb8oS* | ATCACCCGTTCTTCCACTGC | AGGTTTGTGCCCTTTCCGTT |

start site (TSS). We used ENCODE ChIP-Seq Significance Tool (*Auerbach et al., 2013*) to obtain putative transcription factor binding, including identification of genes bound by GCN5/KAT2A, to confirm the identity of putative Kat2a targets.

## Single-cell RNA sequencing and data analysis

Terminal BM samples from *Kat2a* WT and *Kat2a* KO MLL-AF9 primary leukemia animals were collected (WT - 5; KO - 4) and the individual cell samples stored at −150˚C. Cells were thawed and pooled for library preparation. Specifically, 12K live cells per genotype pool were sorted on an Influx sorter (BD) on the basis of YFP expression (reporting *MLL-AF9*), Hoechst 32258 exclusion (live cells) and singlet configuration (pulse width) and used for library preparation with Chromium Next GEM Single Cell 3'GEM, Library and Gel Bead Kit v1 (10XGenomics) aiming at 6K single cells per sample. Library preparation and single-end sequencing on a NextSeq 500 sequencer were done at CRUK Cambridge Research Institute. Raw single cell RNAseq fastq reads were analysed using Cellranger software (v2.1.1) to obtain the cell-gene count-matrix. Seurat (*Butler et al., 2018*) was used for preprocessing the count-matrix data and obtaining differential gene expression between the two genotypes. We employed pairwise distance between gene correlations as a measure of cellular heterogeneity as described (*Mohammed et al., 2017*), by identifying the top 500 highly variable genes in both *Kat2a* WT and KO genotypes based on distance-to-median (DM) and calculating Spearman correlation coefficients between all gene pairs. The correlation matrix was used to compute the pairwise distance measure. RaceID/StemID (*Grün et al., 2016*) algorithms were used for clustering using t-SNE and obtaining pseudo-temporal arrangement of clusters based on entropy information and cluster stem scores. Monocle version 2.6.4 (*Trapnell et al., 2014*) was used as an alternative pseudo-time analysis for *Kat2a* WT and *Kat2a* KO cells. The relative cell state ordering in Monocle was unsupervised, with leukemia self-renewal *Hoxa9* expression employed to determine the directionality of the trajectory. Parameters for the stochastic gene expression were fitted to the two-state promoter model using the D3E algorithm (*Delmans and Hemberg, 2016*) with the Bayesian method option for model fitting. Global normalized data were used in the Robust gene set analysis; cluster-specific parameter derivation included computation of cluster-specific normalization. A multiple linear regression of CV and average gene expression with respect to burst size and burst frequency for *Kat2a* WT and *Kat2a* KO cells were performed using lm() function in R statistical package. P-values for significant coefficients were calculated as a output of the lm() function. Scripts for integration of single-cell RNA-seq and ChIP-seq data were coded in R Language (version 3.4.4) and are provided with this submission. Gene Ontology analysis was performed with the PANTHER online tool (*Mi et al., 2019*), selecting binomial analysis with Bonferroni correction. Gene sets were obtained from the Molecular Signatures Database (MSigDB) (*Subramanian et al., 2005*) to plot gene signatures on tSNE plots. The self-renewal gene signature associated with MLL leukemia, GCM_MLL, was employed in *Figure 4—figure supplement 1A*. For *Figures 6A* and *4* gene sets (MORF_EIF4E, MORF_EIF3S2, MORF_EIF4A2 and MORF_EIF3S6) were pooled to represent a translation-associated gene signature.

## Statistical analysis

Statistical tests performed are specified in the figure legends. Differences were obtained with significant p-value<0.05. Analyses were performed in statistical language R (version 3.4.4) or using Prism version 8.1.2 (GraphPad).

## Acknowledgements

The *Kat2a (Gcn5) Flox* allele mouse strain was a kind gift from Prof Sharon Dent, MD Anderson Cancer Centre, Smithville, TX, USA. The authors are grateful to Dr Matt Wayland for help with Cell Ranger installation and initial processing of the 10X Genomics data; to the Flow Cytometry Facility at the Cambridge Institute for Medical Research, Cambridge for expert assistance with cell sorting; to the Genomics Core Facility at the MRC/WT Cambridge Stem Cell Institute for ChIP-Seq library preparation; to the CRUK Cambridge Research Institute Genomics Core Facility for 10X Genomics library preparation and Next-Generation Sequencing services; and to the Cambridge Biomedical Services for animal husbandry. This work was funded by a Kay Kendall Leukaemia Fund Intermediate Fellowship (KKL888) and by a Leuka John Goldman Fellowship for Future Science (2017) to CP. SP is

funded through a Cambridge-DBT Lectureship; RK was funded by an Isaac Newton Trust (INT) Research Grant and a Wellcome Trust ISSF/INT/University of Cambridge Joint Research Grant to CP; SG is funded by a Lady Tata Memorial Trust PhD Studentship, a Trinity Henry Barlow Trust Scholarship, and the Cambridge Trust; LA is funded by a Rosetrees Trust PhD studentship; KZ received funding from AIRC (Italian Association for Cancer Research) and is the current recipient of a European Commission Horizon 2020 Marie Sklodowska Curie Post-Doctoral Fellowship.

## Additional information

### Funding

| Funder | Grant reference number | Author |
|---|---|---|
| Kay Kendall Leukaemia Fund | KKL888 | Cristina Pina |
| Lady Tata Memorial Trust | PhD studentship | Shikha Gupta |
| Rosetrees Trust | PhD studentship | Liliana Arede |
| Isaac Newton Trust | | Cristina Pina |
| H2020 Marie Skłodowska-Curie Actions | 800274 | Keti Zeka |
| Wellcome | University of Cambridge ISSF | Cristina Pina |
| Wellcome | Cambridge/DBT Lectureship | Sudhakaran Prabakaran |
| Associazione Italiana per la Ricerca sul Cancro | | Keti Zeka |
| Leuka | John Goldman Fellowship for Future Science | Cristina Pina |
| Trinity College, University of Cambridge | Henry-Barlow Trust Scholarship | Shikha Gupta |
| Cambridge Commonwealth, European & International Trust | | Shikha Gupta |

The funders had no role in study design, data collection and interpretation, or the decision to submit the work for publication.

### Author contributions

Ana Filipa Domingues, Formal analysis, Investigation, Visualization, Methodology; Rashmi Kulkarni, Data curation, Software, Formal analysis, Investigation, Visualization, Methodology; George Giotopoulos, Ayona Johns, Shengjiang Tan, Investigation, Methodology; Shikha Gupta, Formal analysis, Investigation; Laura Vinnenberg, Liliana Arede, Elena Foerner, Mitra Khalili, Rita Romano Adao, Investigation; Keti Zeka, Resources, Investigation; Brian J Huntly, Resources, Data curation, Formal analysis, Investigation, Methodology; Sudhakaran Prabakaran, Resources, Data curation, Formal analysis, Supervision, Funding acquisition, Investigation, Visualization, Methodology; Cristina Pina, Conceptualization, Resources, Formal analysis, Supervision, Funding acquisition, Investigation, Visualization, Methodology, Project administration

### Author ORCIDs

Cristina Pina  https://orcid.org/0000-0002-2575-6301

### Ethics

Animal experimentation: Mice were kept in an SPF animal facility, and all experimental work was carried out under UK Home Office regulations. Animal research was regulated under the Animals (Scientific Procedures) Act 1986 Amendment Regulations 2012 following ethical review by the University of Cambridge Animal Welfare and Ethical Review Body (AWERB).

**Decision letter and Author response**
Decision letter https://doi.org/10.7554/eLife.51754.sa1
Author response https://doi.org/10.7554/eLife.51754.sa2

## Additional files

### Supplementary files

• Supplementary file 1. Summary properties of 10X Genomics single-cell RNA-seq data for *Kat2a* WT *vs.* KO *MLL-AF9* primary leukemia.

• Supplementary file 2. Composition of Robust gene set in single-cell RNA-seq analysis of *Kat2a* WT *vs.* KO *MLL-AF9* primary leukemia.

• Supplementary file 3. PANTHER-based Biological Process Gene Ontology overrepresentation analysis of Robust gene set.

• Supplementary file 4. PANTHER-based Biological Process Gene Ontology overrepresentation analysis of differentially expressed genes in STEM-ID clusters 2, 4 and 7 between *Kat2a* WT *vs.* KO *MLL-AF9* primary leukemia cells.

• Supplementary file 5. ENCODE ChIP-seq Significance Tool analysis of differentially-acetylated promoter peaks in *Kat2a* KO *MLL-AF9* primary leukemia (Kat2a acetylation targets).

• Supplementary file 6. PANTHER-based Biological Process Gene Ontology overrepresentation analysis of Kat2a acetylation targets.

• Supplementary file 7. PANTHER-based Biological Process Gene Ontology overrepresentation analysis of Kat2a acetylation targets with reduced Burst frequency in *Kat2a* KO *MLL-AF9* primary leukemia.

• Transparent reporting form

### Data availability

All single-cell RNAseq data and ChIPseq data were deposited in GEO (SuperSeries GSE118769).

The following dataset was generated:

| Author(s) | Year | Dataset title | Dataset URL | Database and Identifier |
|---|---|---|---|---|
| Kulkarni R, Pina C | 2020 | Loss of Kat2a enhances transcriptional noise and depletes acute myeloid leukemia stem-like cells | https://www.ncbi.nlm.nih.gov/geo/query/acc.cgi?acc=GSE118769 | NCBI Gene Expression Omnibus, GSE118769 |

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
