## [Decision Letter]

**Acceptance summary:**

In this study, Pina and colleagues combined chromatin profiling and single-cell transcriptomics of a conditional KO mouse of Kat2a/Gnc5, a histone acetyltransferase central for promoter activity, to study its role in normal hematopoiesis and leukemia development. They describe that hematopoietic selective Kat2a KO does not significantly affect normal hematopoiesis; however, Kat2a KO impairs MLL-AF9-induced murine acute myeloid leukemia in vitro and in vivo by affecting the maintenance of functional leukemia stem-like cells. The authors show that Kat2a loss impacts transcription factor binding and reduces transcriptional burst frequency in a subset of gene promoters, generating enhanced variability of transcript levels. The authors suggest a new mechanism that destabilizing transcriptional variability modulates self-renewal vs differentiation of leukemia stem-like cells in acute myeloid leukemia. This study places the conceptual framework linking transcriptional variability and chromatin dysregulation to leukemia stem cell function, which will have important implications in different tumors and/or distinct stages of cancer evolution.

**Decision letter after peer review:**

Thank you for submitting your article "Loss of Kat2a enhances transcriptional noise and depletes acute myeloid leukaemia stem like cells" for consideration by *eLife*. Your article has been reviewed by three peer reviewers, one of whom is a member of our Board of Reviewing Editors, and the evaluation has been overseen by a Reviewing Editor and Maureen Murphy as the Senior Editor.

The reviewers have discussed the reviews with one another and the Reviewing Editor has drafted this decision to help you prepare a revised submission.

Summary:

In this manuscript, Pina and colleagues used conditional KO mouse of Kat2a/Gnc5, a histone acetyltransferase for H3K9ac, to study its role in hematopoiesis and MLL-AF9 induced leukemia development. They first find that hematopoietic selective (by Mx1-Cre) pIpC-inducible Kat2a KO does not significantly affect normal hematopoiesis; however, Kat2a KO impairs MLL-AF9-induced murine leukemia in vitro and in vivo by affecting the maintenance of functional leukemia stem cells (LSCs). By chromatin profiling ChIP-seq and scRNA-seq analyses, the authors show that Kat2a loss increases transcriptional burst frequency and impacts TF binding such as Myc and Gabpa. The authors suggest that the above mechanisms underlie the perturbations in self-renewal vs differentiation of LSCs. Kat2a target genes were enriched with protein translation associated genes. Additionally, Kat2a inhibition in human MOLM-13 cells significantly affects polysome formation. The authors conclude that Kat2a controls transcriptional bursting in LSCs, and that destabilization of Kat2a target genes shifts leukemia cell fate from self-renewal to differentiation.

Overall this study provides several interesting findings related to the functional and mechanistic requirement of Kat2a in MLL-AF9 leukemia. This manuscript places the conceptual framework linking transcriptional noise and chromatin mutation to the cancer field. The idea that chromatin disruption, operating via increased transcriptional noise, can destabilize cell states, with the increased heterogeneity contributing to cancer evolution and drug resistance has been a popular one – although with little real data to support it. In this study, the emphasis is more that the chromatin disruption destabilizes the cancer progression "program" by depleting the number of cancer stem cells, which is an interesting twist on the standard concepts. The study concludes by suggesting that the loss of the HAT can strongly drop the translational output of the cell, and in doing so, drive escape from the stem cell compartment.

The mouse genetic experiments were appropriately designed and the results were carefully analyzed. This study provides a nice addition to the existing literature, including several from the authors' group, on the function of Kat2a in regulating transcriptional variability in stem cells (Pina et al., 2012 NCB; Teles et al., 2013; Tzelepis et al., 2016; Moris et al., 2018; etc). The overall findings also support an important role of Kat2a in regulating stem cell self-renewal vs differentiation in different model systems. The authors made other important findings, including the effects of Kat2a KO on transcriptional bursting, TF binding, and polysome formation, that increase our understanding of this important histone acetyltransferase in leukemia development. There are several important questions that need to be addressed, as detailed below, to further strengthen the main conclusions. If the remaining questions can be adequately addressed, all of the reviewers felt that this work will have a strong impact and will be of great interest to the hematopoietic, leukemia, and gene regulation communities.

Essential revisions:

1) In assessing MLL-AF9 leukemia differentiation in vitro, the authors used serial replating of CFC assays (Figure 1). However, the description of phenotypes (e.g. compact, mixed, or dispersed) is somewhat vague and unclear, and it is important to include more quantitative measures such as flow cytometry of surface markers and/or expression of signature genes. Images of representative colonies should be provided.

2) Figure 3: pIpC-induced Kat2a gene deletion might not be 100%. Usually, it is not a big issue for bulk RNA-seq; however, for single cell RNA-seq, this raises the concern of whether or not incomplete deletion contributes to the increased heterogeneity and transcriptional noise. As a quality control, the authors are asked to evaluate the degree of Kat2a gene deletion in various cell clusters.

3) scRNA-seq analysis: Figure 4, it will be helpful to provide information about gene signatures and associated annotations (e.g. by GO and/or GSEA) that separate different cell clusters especially for cluster 7. This is to reveal the role for Kat2a in different cell types. In Figure 4—figure supplement 2C and D: are these effects specific to the computational method used? The authors should try one or two more trajectory plotting methods that involve different assumptions.

4) ChIP-seq analysis: it will be helpful to provide additional justifications on the selection of H3K9ac+ only group as the focus of this study. Based on subsection “ChIP-seq data analysis”, it appears that authors focused on H3K4me3 peaks with H3K9ac alone. The logic seems to be that Kat2a/Gnc5 is mainly responsible for promoter-associated H3K9ac. This decision needs to be better justified. The authors should also justify why peaks with H3K9ac+/H3K27ac+ were excluded. How about H3K9ac+ (regardless H3K27ac+ or H3K27ac-) at enhancers (H3K4me1+)? Figure 5C, the authors should comment why there is a similar burst frequency reduction for Kat2a acetylation targets and non-targets.

5) Analysis of polysome and protein translation: One concern is that the observed polysomal content reduction may be caused by off-target effects of the inhibitor on other HATs or ribosome function. These possibilities should be discussed, and the authors should confirm these results using genetic approaches. More importantly, the MOLM-13 leukemia cell line may not recapitulate LSCs. Since the main conclusion is that Kat2a loss impairs LSC maintenance through impaired transcriptional bursting and translation, the authors should perform the analyses on WT vs Kat2a KO MLL-AF9 leukemia cells. If the cell number is limiting for polysome analysis, the authors may consider measuring protein translation directly by OP-Puro incorporation assay using established protocols (e.g. PMID: 24670665) in LSCs (e.g. L-GMPs) in vivo. If the authors could show the significant effect of protein translation in LSCs in vivo upon Kat2a KO, then these findings and conclusions would be very significant and novel.

---

## [Author Response]

Essential revisions:

*1) In assessing MLL-AF9 leukemia differentiation* in vitro*, the authors used serial replating of CFC assays (Figure 1). However, the description of phenotypes (e.g. compact, mixed, or dispersed) is somewhat vague and unclear, and it is important to include more quantitative measures such as flow cytometry of surface markers and/or expression of signature genes. Images of representative colonies should be provided.*

We agree with the reviewers that the description of colony types was incompletely documented, and thank them for pointing this out. We have included photographs of typical compact-type (revised Figure 1D) and mixed-type (revised Figure 1G) colonies, alongside representative flow cytometry profiles (revised Figure 1E and 1H, respectively). The latter document loss of progenitor-associated marker C-Kit and increase in median fluorescent intensity (MFI) of myeloid-lineage associated marker CD11b/Mac1 in mixed-type colonies. These data are described in subsection “Kat2a loss impairs establishment of MLL-AF9 leukemia”. For independent validation of the morphology and surface marker characteristics of MLL-AF9-transformed colonies, we also include similar photographic and flow cytometry documentation of compact and mixed-type colonies obtained from animals that did not undergo pIpC treatment prior to bone marrow collection (Figure 1—figure supplement 2A, C and Figure 1—figure supplement 2B, D, respectively).

2) Figure 3: pIpC-induced Kat2a gene deletion might not be 100%. Usually, it is not a big issue for bulk RNA-seq; however, for single cell RNA-seq, this raises the concern of whether or not incomplete deletion contributes to the increased heterogeneity and transcriptional noise. As a quality control, the authors are asked to evaluate the degree of Kat2a gene deletion in various cell clusters.

We thank the reviewers for their point, which we agree is an important one, and that we have attempted to clarify and address further both in the text (subsection “Increased transcriptional variability associates with loss of Kat2a KO leukemia stem-like cells”) and in the new Figure 4—figure supplement 1A,D-E, in addition to pre-existing panels in Figure 1—figure supplement 1A, D and Figure 2B. The excision of *Kat2a* allele used in this study generates an in-frame product that joins the first 2 and the last exons (Figure 1—figure supplement 1A), and which is transcribed at levels at least similar to the full-length version of the wild-type gene (Figure 1—figure supplement 1D). Given the 3’-bias of the 10X Genomics procedure we have used for single-cell RNA sequencing in this study, we cannot reliably quantify the level of residual *Kat2a* expression in individual clusters from the sequencing data. We have nevertheless attempted to address this point experimentally in 2 ways.

1) We have measured *Kat2a* expression in primary and secondary MLL-AF9 leukemias from *Kat2a* WT and *Kat2a* KO. We used a Taqman assay (Mm00517402_m1) targeting an amplicon at the boundary between exons 6-7, which is exclusive to the unexcised allele. Reassuringly, we see that the KO expression is more than 95% reduced compared to WT (Figure 2B), and that this level of KO is maintained even amongst the (fewer) secondary KO leukemias generated (Figure 4—figure supplement 1D) suggesting a profound locus excision with minimal contribution from cells escaping locus deletion.

2) Perhaps more importantly, we performed single-cell RT-PCR on double-sorted Lin-Kit+ Sca1-FcgR+ cells obtained from 2 secondary leukemias/genotype generated during this study and from which we had sufficient material. We designed intron-spanning primers against exons 11-13 of *Kat2a*, again targeting a portion of the transcript included in the deletion, and multiplexed them with *Hprt* primers to perform nested single-cell RT-PCR as previously described by us in Pina et al., 2012. We analysed 88 Lin-Kit+ Sca1-FcgR+ cells/genotype and could not detect any *Kat2a* expression, while this was apparent in 79% of WT cells (Figure 4—figure supplement 1E).

Taken together, these data support minimal contribution from unexcised cells to both cellular and molecular analyses, particularly to single-cell RNA seq analysis of the stem-like cell compartments, which we explore in detail.

3) scRNA-seq analysis: Figure 4, it will be helpful to provide information about gene signatures and associated annotations (e.g. by GO and/or GSEA) that separate different cell clusters especially for cluster 7. This is to reveal the role for Kat2a in different cell types. In Figure 4—figure supplement 2C and D: are these effects specific to the computational method used? The authors should try one or two more trajectory plotting methods that involve different assumptions.

We thank the reviewers for raising these points that undoubtedly inform and add detail to our scRNA-seq analysis of *Kat2a* loss.

Concerning the point on cluster-specific gene annotations, we include 2 additional panels as the revised Figure 4—figure supplement 1B and revised Figure 6A, which demonstrate an association between the stem-most clusters 2, 4 and 7 as defined by STEM-ID and gene sets associated with MLL-associated self-renewal (GCM-MLL) – Figure 4—figure supplement 1B – and with translation-associated genes (MORF_EIF4E, MORF_EIF3S2, MORF_EIF4A2 and MORF_EIF3S6) – Figure 6A. These data reinforce the notion of the stem-like nature of the cells in those clusters, as well as their association with translational machinery, which we show in this manuscript to be targeted by *Kat2a* loss. We found cluster-specific gene annotations to be of limited value due to the reduced number of unique cluster-defining genes. However, we did perform analysis of genes differential between WT and KO cells (as per D3E) in each of the clusters. There is considerable overlap in enriched categories between clusters, although clusters 2 and 4 have clearer associations with cell death / apoptosis, which are not present as enriched differential categories in cluster 7. These data have been included as Supplementary file 4, and are referred to in subsection “Increased transcriptional variability associates with loss of Kat2a KO leukemia stem-like cells”.

In response to reviewers’ comments, we also employed Monocle to infer differentiation trajectories in *Kat2a* WT and *Kat2a* KO leukaemias (revised Figure 4—figure supplement 2E and F, respectively, called in subsection “Increased transcriptional variability associates with loss of Kat2a KO leukemia stem-like cells”). Monocle-inferred trajectory of *Kat2a* KO leukemia denotes more extensive branching from the self-renewal arm and disconnection between early and late arms of differentiation compatible with the heterogeneity of fate decisions captured by STEM-ID. Significantly, the same cells are captured by both methods at the start and end of the WT and KO trajectories, reinforcing the robustness of the conclusions.

4) ChIP-seq analysis: it will be helpful to provide additional justifications on the selection of H3K9ac+ only group as the focus of this study. Based on subsection “ChIP-seq data analysis”, it appears that authors focused on H3K4me3 peaks with H3K9ac alone. The logic seems to be that Kat2a/Gnc5 is mainly responsible for promoter-associated H3K9ac. This decision needs to be better justified. The authors should also justify why peaks with H3K9ac+/H3K27ac+ were excluded. How about H3K9ac+ (regardless H3K27ac+ or H3K27ac-) at enhancers (H3K4me1+)? Figure 5C, the authors should comment why there is a similar burst frequency reduction for Kat2a acetylation targets and non-targets.

We note the reviewers’ point and have explained further our focus on H3K4me3+ H3K9ac-only peaks as targeted by Kat2a. The choice was entirely guided by the data and not justified a posteriori by it, as explained in more detail in subsection “Kat2a regulates transcription factor binding and bursting activity of pro”. Kat2a acetylation activity has been associated with H3K9ac of promoters, a finding recently reiterated by Sharon Dent’s group in embryoid bodies (Wang et al., 2018) and ourselves in ES cells (Moris et al., 2018). However, earlier reports suggested that Kat2a might also exert H3K9ac activity at enhancers, specifically in the context of the ATAC acetylation complex (Krebs et al., 2011), and we interrogated both promoters (Figure 5A) and enhancers (revised Figure 5—figure supplement 1E) for Kat2a-dependent (K9) and non-Kat2a-dependent (K27) acetylation marks. Analysis of co-localized peaks indicated that promoter H3K9ac alone was specifically lost upon *Kat2a* KO. The same was not true at H3K4me1+ enhancers, or of H3K27ac, thus directing our focus to promoter H3K9ac-only peaks.

Additionally, we have commented further on the reduction in burst frequency amongst *Kat2a* non-targets (revised Figure 5C) by advancing potential explanations (subsection “Kat2a regulates transcription factor binding and bursting activity of pro”): 1. We have focused on loss of peak detection rather than reduction in peak intensity, making it likely that some Kat2a target regions may have been missed by our analysis and thus considered amongst Kat2a non-target peaks. 2. Also, Figure 5C estimation of burst frequency uses all cells, rather than putatively functionally identical cells according to clusters as in Figure 5D: it is possible that the underlying functional heterogeneity erode differences between genotypes. 3. We have previously shown that transcriptional noise is propagated through gene regulatory networks to downstream target genes that are not directly regulated by Kat2a (Moris et al., 2018). The fact that the analysis in revised Figure 5C uses all the cells makes it more likely that downstream regulatory connections are captured in the data and may contribute to differences in burst frequency between genotypes. 4. Finally, we cannot exclude that loss of *Kat2a* impact other histone lysine acylations with transcriptional consequences similar to those of acetylation: a recent example is lysine succinylation, which has not been explored globally or in the context of leukaemia, and is an interesting research avenue in the future. This latter point is highlighted in the Discussion section.

*5) Analysis of polysome and protein translation: One concern is that the observed polysomal content reduction may be caused by off-target effects of the inhibitor on other HATs or ribosome function. These possibilities should be discussed, and the authors should confirm these results using genetic approaches. More importantly, the MOLM-13 leukemia cell line may not recapitulate LSCs. Since the main conclusion is that Kat2a loss impairs LSC maintenance through impaired transcriptional bursting and translation, the authors should perform the analyses on WT vs Kat2a KO MLL-AF9 leukemia cells. If the cell number is limiting for polysome analysis, the authors may consider measuring protein translation directly by OP-Puro incorporation assay using established protocols (e.g. PMID: 24670665) in LSCs (e.g. L-GMPs)* in vivo*. If the authors could show the significant effect of protein translation in LSCs* in vivo *upon Kat2a KO, then these findings and conclusions would be very significant and novel.*

We thank the reviewers for their comments, which have indeed strengthened our findings. We have performed OP-Puro incorporation experiments ex vivo using cells isolated from the spleens of MLL-AF9 *Kat2a* WT and KO secondary leukemias produced by this study. We have included the results in revised Figure 6C-D, with the data reported in subsection “Kat2a participates in AML maintenance through regulation of translation-associated 1 genes “and discussed in subsection “Kat2a participates in AML maintenance through regulation of translation-associated 1 genes”. We observed a reduction in OP-Puro incorporation, that was both of fraction of incorporating (i.e. actively translating) cells and the amount incorporated (i.e. level of activity) in *Kat2a* KO leukemia cells with an L-GMP phenotype. Incidentally, we have also made similar observations in another AML model upon *Kat2a* KO (data not shown), suggesting a specific Kat2a-mediated effect worth exploring in the future.